# Nanopore sequencing reveals that DNA replication compartmentalisation dictates genome stability and instability in *Trypanosoma brucei*

Marija Krasiļņikova [1], Catarina A. Marques [1] ✉, Emma M. Briggs [1,2,3], Craig Lapsley[1], Graham Hamilton [4], Dario Beraldi [1], Kathryn Crouch [1] & Richard McCulloch [1] ✉

The *Trypanosoma brucei* genome is structurally complex. Eleven megabase-sized chromosomes each comprise a transcribed core flanked by silent subtelomeres, housing thousands of *Variant Surface Glycoprotein* (*VSG*) genes. Additionally, hundreds of sub-megabase chromosomes contain 177 bp repeats of unknown function, and *VSG* transcription sites localise to many telomeres. DNA replication dynamics have only been described in the megabase chromosome cores, and in the single active *VSG* transcription site. Using a Nanopore genome assembly, we show that megabase chromosome subtelomeres display a paucity of replication initiation events relative to the core, correlating with increased instability. In addition, replication of the active *VSG* transcription site is shown to originate from the telomere, likely causing targeted *VSG* recombination. Lastly, we provide evidence that the 177 bp repeats act as conserved DNA replication origins, explaining submegabase chromosome stability. Compartmentalized DNA replication therefore explains how *T. brucei* balances stable genome transmission with localised instability driving immune evasion.

The fullest possible understanding of the genome sequence is a critical resource to describe and analyse the biology of an organism, including of genome transmission and stability through DNA replication. The genome of the single-celled eukaryotic parasite, *Trypanosoma brucei*, is shaped by processes that govern its survival during host infection. The parasite's modest ~60 Mb (diploid) genome[1] is mainly found on 11 'megabase' chromosomes (~1–5 Mb size range), where virtually all genes are expressed, unusually, from a small number of multigenic transcription units that reside in the diploid chromosome 'cores'[2].

However, further aspects of the genome, occupying possibly 50% of its total content, are largely devoted to the process of antigenic variation, in which the trypanosome continually changes the identity of its Variant Surface Glycoprotein (VSG) 'coat' during mammalian host infection[3,4]. Only one VSG is expressed in a cell at one time, from one of ~15 bloodstream *VSG* expression sites (*VSG* BESs), which are transcription loci immediately adjacent to the telomeres[5]. Antigenic variation relies on a silent archive of >2000 *VSG* genes, most of which are found in arrays within the subtelomeres of the megabase

¹University of Glasgow Centre for Parasitology, The Wellcome Centre for Integrative Parasitology, University of Glasgow, School of Infection and Immunity, Sir Graeme Davies Building, 120 University Place, Glasgow G12 8TA, United Kingdom. ²University of Edinburgh, Institute for Immunology and Infection Research, School of Biological Sciences, Edinburgh, United Kingdom. ³Biosciences Institute, Cookson Building, Newcastle University, Framlington Place, Newcastle upon Tyne NE2 4HH, United Kingdom. ⁴MVLS Research Facilities, University of Glasgow, Wolfson Wohl Cancer Research Centre, Garscube Estate, Switchback Rd, Bearsden, Glasgow G61 1QH, United Kingdom. ✉e-mail: Catarina.DeAlmeidaMarques@glasgow.ac.uk; Richard.mcculloch@glasgow.ac.uk

chromosomes, thus constituting a very large, transcriptionally silent genome compartment positioned between the chromosome cores and telomeric *VSG* BESs[1,6,7]. These *VSG*-rich subtelomeres are notably variable between strains and subspecies of *T. brucei*[8], to the extent that *VSG* content is not equivalent between chromosome homologues[9]. Beyond the megabase chromosomes, *T. brucei* has evolved a large number of sub-megabase chromosomes, named mini- (~50-150 kb in size) and intermediate-chromosomes (150–700 kb). These chromosomes appear to house only telomere-proximal silent *VSGs* or *VSG* BES, thus expanding the *VSG* archive[10]. The main sequence feature of these mitotically stable sub-megabase chromosomes[11] is 177 bp repeats[10], but their function and how they may have evolved is unclear.

Our understanding of DNA replication programming in *T. brucei*, though growing, is far from complete[12]. Marker Frequency Analysis via Illumina sequencing (MFA-seq; termed sort-seq in yeast)[13] has described multiple sites of DNA replication initiation, termed origins, in the megabase chromosome cores[14]. Each origin co-localises with the binding of at least one subunit of the Origin Recognition Complex (ORC) at the boundaries of some of the multigene transcription units, and origin location appears invariant between strains and in at least two life cycle stages[14,15]. However, at the time it was not possible to accurately map MFA-seq data or ORC binding to the megabase chromosome subtelomeres[14,16], as these were poorly assembled. Further work, analysing patterns of DNA replication through labelling single DNA molecules detected DNA replication forks emanating from one subtelomere of chromosome 1, but where they initiated from was not resolved[17]. A similar approach inferred greater numbers of origins than were mapped by MFA-seq, but as the labelled molecules were not positioned in the genome, the locations of any presumptive 'extra' origins are unknown[18]. MFA-seq has shown that the single actively transcribed *VSG* BES is distinct from all silent *VSG* BESs in being replicated early in S-phase of bloodstream form cells, but the origin and direction of such replication is unknown[15,19]. Finally, no data has mapped DNA replication in the sub-megabase chromosomes. Here, we sought to address these limitations, using MFA-seq to describe DNA replication dynamics in a more complete assembly of the *T. brucei* genome.

Prior to 2018, *T. brucei* genome assemblies reflected the limitations of short-read sequencing[1]: whereas the cores were accurately assembled, the subtelomeres were incompletely assembled and not linked to chromosome cores, and little information was available for the sub-megabase chromosomes. Sequencing of the *VSG* BESs was achieved through targeted cloning in yeast[20] but this approach did not link the transcription sites to the megabase chromosomes. PacBio long-read DNA sequencing[21] aided by Hi-C DNA interaction data[9] greatly improved this picture: subtelomeric sequences have been near fully assembled and assigned to specific chromosome cores; many *VSG* BESs have been assigned to chromosomes; distinct allelic chromosomes have been resolved; and, in one strain, some mini- and intermediate chromosomes have been assembled. However, some features remain incompletely examined: the large centromeres have not been fully sequenced; 260 relatively short sequences (1–142 kb in length, accounting for 14.8% of the reference genome), termed 'unitigs', remain unassigned to the genome; and sequences connecting the megabase subtelomeres and cores, as well as subtelomeres and *VSG* BESs are unexplored[4]. Each of these issues limits understanding of *T. brucei* DNA replication.

Here, we decided to ask if Oxford Nanopore Technologies sequencing might complement Müller et al.'s PacBio and Hi-C assembly of the *T. brucei* Lister 427 genome[9] and help us resolve unanswered questions about DNA replication dynamics. The utility of long-read Nanopore sequencing for genome assembly has recently been demonstrated in *T. cruzi*, where the presence of many multigene families and abundant transposable elements proved to be a considerable impediment to assembly[22,23], while it has also allowed telomere-to-telomere chromosome assemblies of several other genomes[24,25]. One notable feature of Nanopore sequencing is the generation of ultra-long (hundreds of kb) reads, which we reasoned in *T. brucei* may allow increased understanding of repetitive regions in the genome, amongst which the very long centromeres[9,26,27], estimated to be 20–120 kb in size[28], are very early-acting origins[14]. In addition, traversing the 177 bp and 50 bp repeats[29] could reveal how the sub-megabase chromosomes and *VSG* BES are replicated, respectively.

Using Nanopore sequencing, we performed de novo assembly of the *T. brucei* Lister 427 genome, allowing megabase chromosome cores and subtelomeres to be connected, *VSG* BESs to be connected upstream to subtelomeres and downstream to telomeres, sub-megabase chromosomes to be assembled, and complete sequences of centromeres to be obtained. The assembly revealed that several megabase centromeres share the 177 bp repeats found in the sub-megabase chromosomes, and that a number of these small chromosomes do not only harbour silent *VSGs* or *VSG BESs*, but also numerous transcribed genes, indicating they can be more than simply *VSG* archives. MFA-seq analysis using the Nanopore assembly revealed highly compartmentalised DNA replication and stability across the *T. brucei* genome. First, we show that there is a pronounced difference in both DNA replication dynamics and stability between the core and subtelomeres of the megabase chromosomes, explicable by origin number. Second, we demonstrate that life cycle stage-specific early DNA replication of the single active *VSG* BES initiates from the telomere, providing an explanation for targeted *VSG* recombination during antigenic variation. Finally, we reveal that the 177-bp repeats are previously undetected, sequence-conserved origins of DNA replication that provide a means for stable transmission of all classes of *T. brucei* chromosomes.

## Results
### Nanopore sequencing and assembly of the *T. brucei* Lister 427 genome
From the combined use of MinION and Flongle R9.4.1 flow cells, 319,075 reads, totalling 3.56 Gb of genomic DNA sequence, were generated from bloodstream form (BSF) *Trypanosoma brucei brucei* Lister 427 cells (Supplementary Table 1). Mean and median read lengths were 11,152 and 4307 bp, respectively, though these metrics were influenced by abundant reads of ~1 kb, mainly derived from mitochondrial minicircles. Maximum read length was 345,688 bp. The resulting reads were used for de novo genome assembly using Canu, including read correction and trimming[30]. Additional sequence correction, following assembly, was carried out by four iterations of Pilon[31] using Illumina paired-end (2 x 75 bp) reads generated from the same *T. brucei* strain. Relative to the PacBio and Hi-C assembly (hereafter called the Müller genome)[9], the Nanopore assembly contained an additional 5.5 Mb of genome sequence, and contig number was reduced from 317 to 166 (Table S2). Nonetheless, genome completeness measured using BUSCO[32] indicated the Nanopore assembly was comparable to that of the Müller genome (Supplementary Table 2), and similar numbers of predicted *VSG* genes were found (Supplementary Table 2). Only 72 putative genes and pseudogenes (Supplementary data 1) were predicted in the Nanopore assembly that were not detected in the Muller genome, and all were present in megabase chromosome contigs.

13 contigs in the Nanopore assembly were >1 Mb in size, with the longest ~5 Mb (Fig. S1). Seven of these contigs encompassed full-length centromeres, and nine included both core and subtelomere compartments of the genome (Fig. S1). Collectively, these contigs from the Nanopore assembly improved genome contiguity across genomic compartments, thus allowing us to substantially extend our understanding of DNA replication dynamics in *T. brucei*, as outlined below.

### Compartmentalised DNA replication in the *T. brucei* megabase chromosomes
For 10 of the 11 megabase chromosomes, the Nanopore assembly generated contigs that bridged at least two previously separate contigs (Table S3; Fig. 1A and Fig. S2 provide examples of these contigs); only

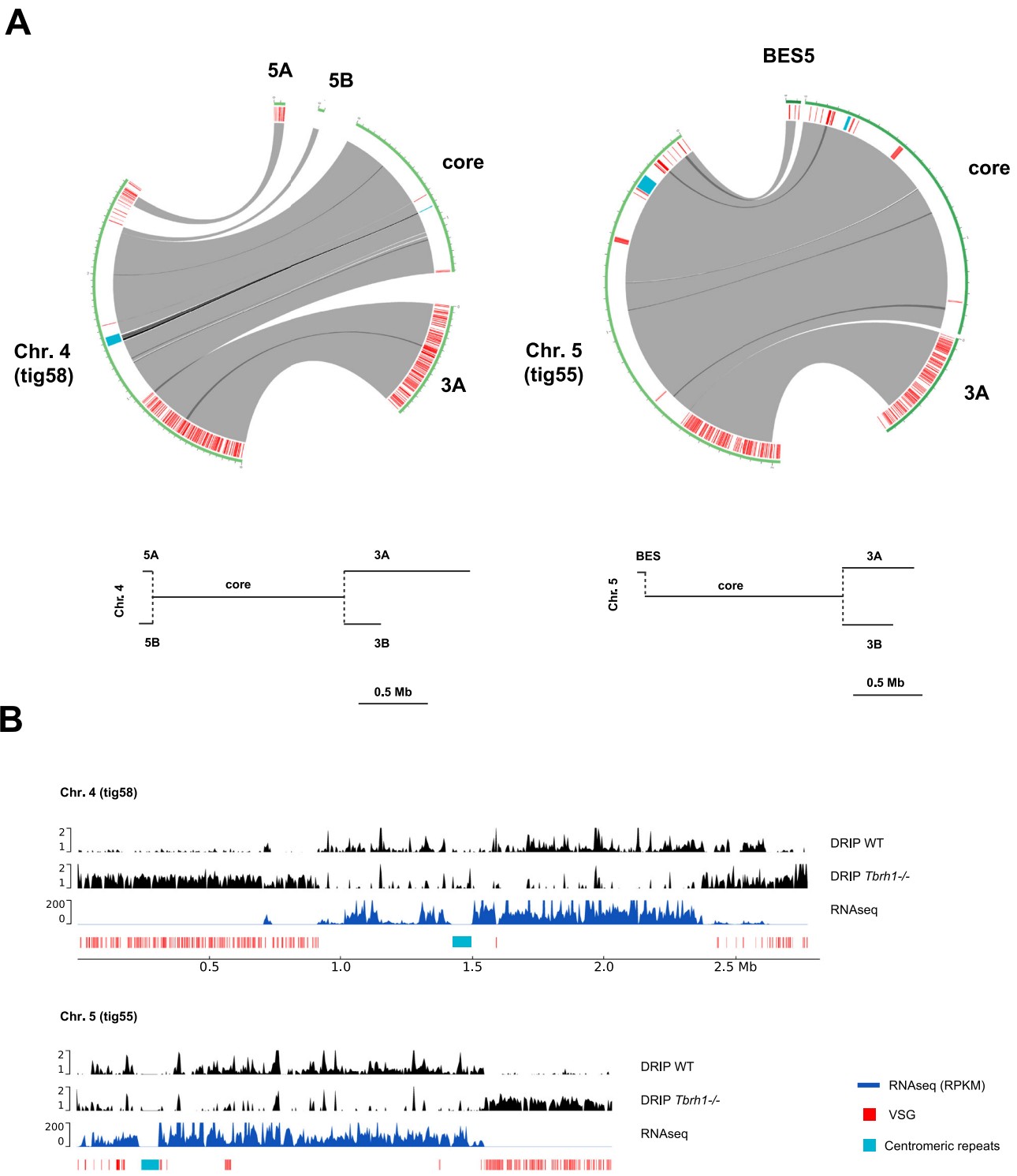

**Fig. 1 | Connecting the cores, subtelomeres and bloodstream VSG expression sites of the *T. brucei* megabase chromosomes through Nanopore long-read sequencing. A** Circos plots highlighting synteny between the Muller genome assembly and Nanopore contigs (tigs) of two megabases chromosomes (4 and 5). Grey ribbons represent overlaps, black ribbons represent multiple overlaps within a region, red denotes *VSG* genes, and blue denotes a centromere; for clarity of comparison, the organisation of the transcribed core, transcriptionally silent sub-telomeres (numbered 3 A, 3B, 5 A, 5B) and bloodstream VSG expression sites (BESs)

in these chromosomes in the Muller genome assembly are diagrammed in the lower panel (adapted from[9]). **B** Distinct behaviour of the core and subtelomere compartments of chromosomes 4 and 5 Nanopore contigs is shown by mapping of R-loops (DNA-RNA hybrid immunoprecipitation and sequencing, DRIP; data shown as IP/input) in wild type (WT) bloodstream from cells and RNase H1 null (-/-) mutants, as well as by mapping of RNA-seq data from WT cells (RPKM: reads per Kb per million mapped reads). Elements of panel A were created in BioRender. McCulloch, R. (2025) https://BioRender.com/u96k590.

in chromosome 2 was such bridging not observed. Potentially complete chromosomes encompassing subtelomere-core-subtelomere, or subtelomere-core-*VSG* BES were also found. Overall, there was good correspondence with the Muller genome, though some differences were apparent: connections between the core and 'A' and 'B' subtelomeres of some chromosomes differed, and synteny was sometimes interrupted (e.g. by incorporation of previously unassigned unitigs into Nanopore contigs). Importantly for understanding DNA replication (see below), centromeres were expanded in most chromosomes by Nanopore sequencing.

By connecting a number of previously unconnected core and subtelomere contigs, the Nanopore assembly allowed scrutiny of the regions between the highly transcribed core and the mainly transcriptionally silent subtelomeres[9,33]. Perhaps unexpectedly, no consistent sequence was detected at the boundary of these genomic compartments. Despite this, the behaviour of the core and subtelomere compartments is highly distinct. One illustration of the sharp divide between the compartments is illustrated by mapping DNA-RNA hybrids from wild type cells and RNase H1 null mutants[34,35] to the Nanopore assembly (Fig. 1B, Supplementary Fig. 3A). In wild type cells, clear mapping of the hybrids was seen across the transcribed cores, but more limited mapping was seen in the mainly transcriptionally silent subtelomeres, whereas after loss of RNase H1, RNA-DNA hybrids became more abundant in the subtelomeres than in the core. In addition to this epigenetic feature, analysis of base composition revealed a stark and immediate reduction in GC content as the cores transitioned to subtelomeres (Supplementary Fig. 4). Hence, despite the lack of a clear boundary feature, RNA-DNA hybrid distribution, base composition and differential transcription[9] all indicate differences in behaviour between the two linked compartments of the megabase chromosomes.

Genome-wide DNA replication dynamics in *T. brucei* has only been examined to date using MFA-seq[36,37], a technique where DNA read depth across each chromosome is compared in replicating cells (early S-phase or late S-phase) relative to non-replicating cells (G2M or G1)[14,19]. However, MFA-seq mapping was only possible across the transcribed megabase chromosome cores, due to insufficient assembly of the subtelomeres. Figure 2 shows MFA-seq analysis of the >1 Mb Nanopore contigs, for both bloodstream (mammalian stage) and procyclic (insect stage) form cells, thereby allowing a comparison of DNA replication dynamics in the subtelomeres and cores for most megabase chromosomes. Fig. S5 shows the same data mapped to the Müller genome[9]. These analyses revealed a dearth of detectable DNA replication initiation in the subtelomeres: other than around the subtelomeric centromeres on chromosomes 9, 10 and 11, there were no clearly detectable MFA-seq peaks in the *T. brucei* chromosome subtelomeres (Fig. 2 and Supplementary Fig. 5). To quantify the relative number of predicted origins in the subtelomeres and cores, Fig. 3A shows Circos plots of the MFA-seq data mapped against the Müller genome, release v46 (tritrypDB.org), where the core and subtelomere compartments are assembled separately, thus allowing for easier comparative visualisation: whereas 47 MFA-seq origins were seen in the ~23 Mb core genome[19], only 6 MFA-seq origins (all centromeric) were predicted in the ~19 Mb subtelomeric genome compartment. Thus, the compartmentalisation of the megabase chromosomes is not limited to differing transcription, RNA-DNA hybrid levels and GC content between the cores and subtelomeres but extends to differing dynamics of DNA replication.

MFA-seq revealed that centromeres located in the subtelomeres of chromosomes 9, 10 and 11 always displayed a peak, indicating they contain an origin or origins (Figs. 2, 3A). To ask if these subtelomeric origins initiate DNA replication as efficiently as centromeres in the core, we generated metaplots of MFA-seq signal across all the centromeres (Supplementary Fig. 6). To do this, we identified all potential centromeric repeat regions in the Nanopore assembly by initially

generating a blastn database containing flanking sequences surrounding gaps in the annotated centromeres from the Müller genome[9], and then using blastn to localise matching sequences in the Nanopore assembly. The results were further manually refined based on sequence composition, synteny and Tandem Repeats Finder analyses. In total, 23 centromeric repeat candidates were retrieved (Table S4), nine of which are likely full-length, based on the presence of flanking regions. The length of the centromeres varied considerably (30.2–105.2 kb, full length; 6.1–95.7 kb, partial), but the predicted large sizes and high AT content were comparable to previous estimates from *T. brucei* strain TREU927[26,28]. Sequence content of the Nanopore-assembled centromeres was very variable but, nonetheless, strikingly similar amplitude and width of the MFA-seq peaks was seen across all the assembled centromeres (Fig. S6). These data show that localisation of centromeres in the largely transcriptionally silent subtelomeres does not affect the use of these genome features as origins relative to when they are localised in the highly transcribed megabase chromosome core.

## Compartmentalised genome stability in the *T. brucei* megabase chromosomes

Recent chromatin capture analyses in archaeal *Sulfolobus* species have indicated that these organisms' circular genome is compartmentalised into two domains, which display differing levels of gene expression[38]. The three *Sulfolobus* origins localise to the more transcriptionally active domain and, furthermore, the mutation rate is greater in the less active, origin-free domain and with greater distance from an origin[39]. Given these observations, we asked if the *T. brucei* genome's arguably more extreme compartmentalisation between the transcribed, origin-rich core and largely untranscribed, origin-sparse subtelomeres might also extend to stability. To do so, we sub-cloned wild type, *rad51* null mutant[40] and *brca2* null mutant[41,42] bloodstream form *T. brucei* cells and grew two clones of each for 23 passages (approximately 140 population doublings). DNA from each starting population, the clones prior to passage, and the passaged clone populations was Illumina-sequenced and mapped to the Müller genome (again, because separate assembly of the core and subtelomere compartments allows for easier comparative visualisation; tritrypdb.org, genome release v46). Quantification of read depth before and after subcloning/passaging revealed reduced mapping to the subtelomeric regions relative to the core in all cell types, and that loss of subtelomeric read mapping was most extensive in the *rad51* mutants (Supplementary Fig. 7). Comparing read depth in the two clones of each cell type at the end of the growth experiment relative to the start revealed compartmentalised instability across the time course (Fig. 3B): in all cell types, more regions of reduced read depth were seen in the subtelomeres than the core genome; in addition, the extent of subtelomeric mapping loss (both depth of read mapping, and number of regions with reduced mapping) was increased in both the *rad51* and *brca2* null mutants. Thus, the compartmentalisation of DNA replication between the cores and subtelomeres of the megabase chromosomes is associated with differing levels of stability in the two compartments and is influenced by homologous recombination.

## DNA replication at the telomeres of bloodstream-form *T. brucei*

Transformation-associated recombination (TAR) cloning in yeast, using the *VSG* BES promoter as a recombination target[20], allowed the sequencing of 14 distinct *VSG* BESs in the Lister 427 genome[5]. In this Nanopore genome assembly, contigs containing 15 *VSG* BES sequences could be discerned (Table S5), three of which were duplicated. 10 of these *VSG* BES contigs contained full-length 50 bp repeat regions found upstream of *VSG* BESs[29], as well as subtelomere sequence, and thus provide genomic context for the transcription sites. Surprisingly, only three of the contigs contained telomere sequence (Table S5). The explanation for this lack of completeness appears to reside in

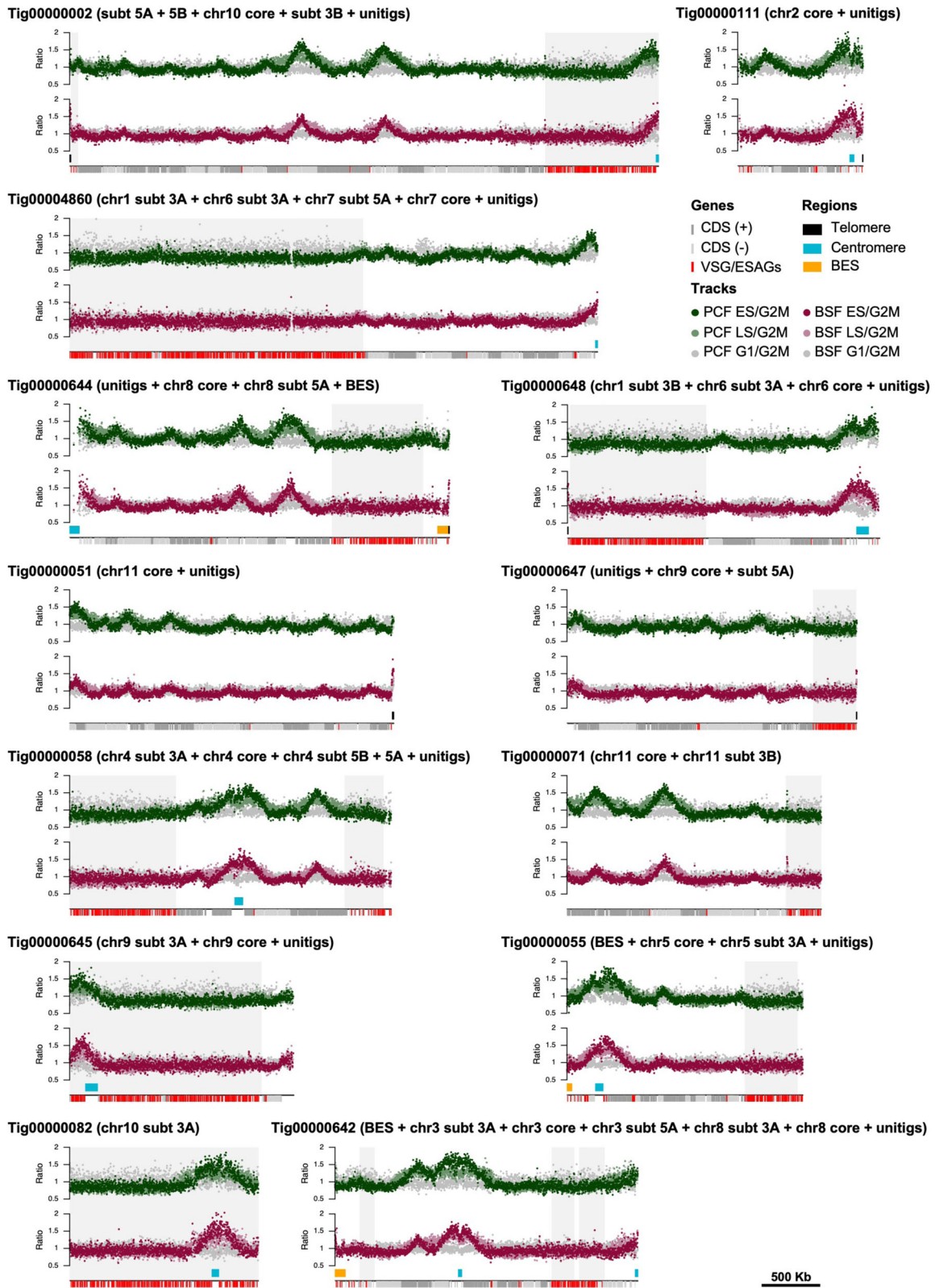

**Fig. 2 | Compartmentalisation of DNA replication between the cores and subtelomeres of the _T. brucei_ megabase chromosomes.** DNA replication dynamics are shown across 13 >1 Mb Nanopore contigs (tig) of the megabase chromosomes as MFA-seq ratios (y-axis), comparing early S phase/G2-M phase cells (dark red, bloodstream cells, BSF; dark green, procyclic form, PCF), late S/G2M cells (light red, BSF; light green, PCF), and G1/G2M cells (grey for both life cycle stages); each point represents a 1 kb bin. For each contig, the corresponding chromosome core and subtelomere (regions shaded in light grey) identified in the Müller genome are indicated, as is whether the contig incorporates unitig sequences. Directional gene clusters in the core (dark grey and light grey depicting the different transcribed strands), centromeres (blue), and _VSGs_ (red) within the subtelomeres are indicated. All graphs are scaled according to contig size. Source data are provided as a Source Data file.

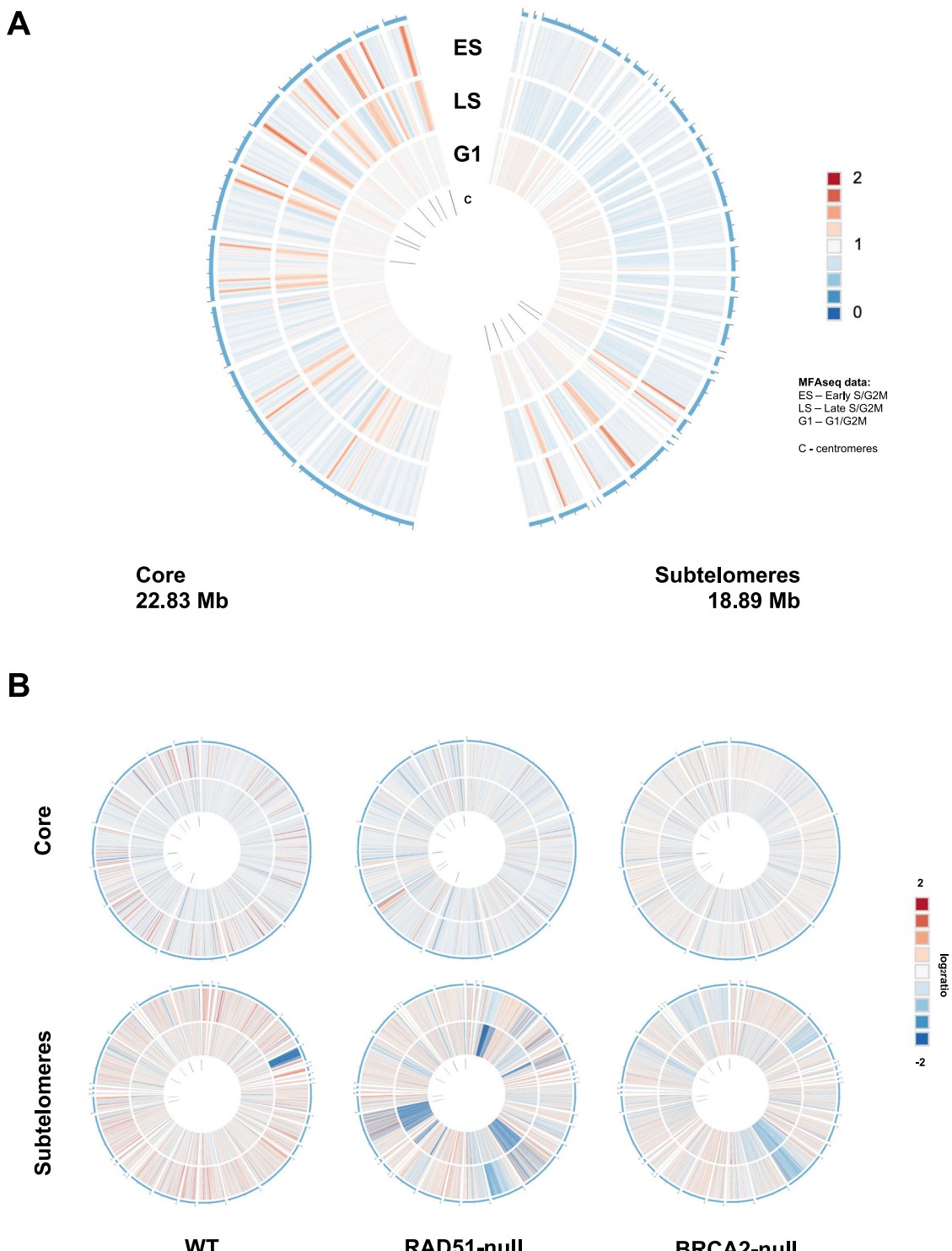

**Fig. 3 | Compartmentalisation of DNA replication and stability between the cores and subtelomeres of the *T. brucei* megabase chromosomes. A** MFA-seq mapping to the core (left) and subtelomeric (right) compartments of the 11 megabase chromosomes; total sequence length of each compartment is indicated. Read depth ratios of early S-phase (ES), late S-phase (LS) and G1-phase cells (all BSF) are shown as heat maps relative to G2-M cells. Locations of centromeres are indicated. **B** Change in read depth mapping across the core and subtelomeric compartments of two clones (separate circles) of wild type (WT), RAD51 null mutant and BRCA2 null mutant cells after 23 passages of growth in culture; data are plotted as log2 ratios and locations of centromeres are indicated.

*VSG*-proximal sequences, since seven of the 15 contigs terminated within the 70 bp repeats and three terminated within the *VSG*-adjacent 'co-transposed region'[43]. Because we were able to assemble 50 bp repeats, 177 bp repeats (see below) and the very large centromeres from Nanopore sequencing data, it is unlikely that the 70 bp repeats

represent a unique assembly challenge. Analysis of the base quality of reads spanning the 70 bp repeats did not suggest they are especially problematic for the Nanopore sequencing process itself, but any sequences that contain telomere repeats were of notably lower quality (Supplementary Fig. 8). Thus, it may be that features of the *T. brucei*

telomere impact Nanopore sequencing, and thereby impede assembly of sequences in their proximity, including BESs.

Using the TAR sequences[5] as a reference, we previously reported that in *T. brucei* bloodstream form cells the active *VSG* BES replicates early in S phase, while all silent BESs are very late replicating[19]. However, we could not determine the direction or initiation site of the DNA replication within or around the any of these loci[19]. Since the Nanopore assembly extends at least some *VSG* BESs upstream to the 50 bp repeats and beyond, and downstream to include the telomere repeats, we mapped the MFA-seq data to all *VSG* BES-containing contigs (Fig. 4). This analysis revealed several facets of telomere-proximal DNA replication dynamics in *T. brucei*. First, we confirmed our previous observation that the active *VSG* BES, BES1 (here, in two copies; Fig. 4A, tig652 and tig653), is the sole early replicating *VSG* transcription site in these bloodstream form (BSF) cells, since only in these contigs was intra-BES S/G2M ratio >1. The data also confirmed that BES1 is a late-replicated region of the genome in procyclic form (PCF) cells, where it is transcriptionally silent, since in these cells intra-BES S/G2M ratios were ~1 (Fig. 4A). Second, the new contigs ruled out the possibility that early replication of BES1 (and indeed any *VSG* BES-proximal replication) results from upstream initiation. In tig652 BES1 is incomplete, but it is connected to the subtelomeric sequence upstream of the 50 repeats. In the subtelomeric sequence of this contig, mean BSF S/G2M ratios were notably lower than those observed within the BES1 transcribed region of both tig 652 and tig653 (Fig. 4A). Moreover, subtelomeric S/G2M ratios upstream of BES1 were comparable to those seen both upstream of and within the BES sequences of contigs containing silent *VSG* BESs, in both BSF and PCF cells (Fig. 4B). These data appear to rule out the 50 bp repeats acting as a replication origin. Third, the Nanopore assembly revealed life cycle stage-specific telomere-directed DNA replication. Tig653 encompasses all of BES1, including the telomere, but lacks a complete 50 bp repeat track or upstream subtelomeric sequence (Fig. 4A, Supplementary Table 5). Here, BSF S/G2M ratio was >1 throughout the BES but was most marked in the telomere tract and appeared to diminish with distance from the chromosome end (Fig. 4A). In contrast, whilst there was no evidence of mean BSF S/G2M ratios >1 within the BES sequence of any silent *VSG* BES contig (Fig. 4B), the telomere tract of both tig137 (silent BES10) and tig644 (silent BES8) displayed >1 S/G2M ratios in BSF cells but not in PCF cells (Fig. 4B). Furthermore, the telomere tract of contigs representing the ends of chromosomes that do not harbour *VSG* BESs also showed S/G2M ratios >1 in BSF but not PCF cells, and the BSF telomere signals were markedly greater than in connected subtelomere or core sequence (Fig. 4C). To examine these observations further, we mapped the MFA-seq data to any Nanopore contig with telomeric TTAGGG repeats at the contig end (Fig. 4D). In BSF cells, there was clear enrichment of MFA-seq signal at all telomeric repeats, an effect most noticeable in the early S/G2M data. Most of this MFA-seq signal did not clearly extend much into sequences upstream of the telomeres and, importantly, equivalent S/G2M signal enrichment was not seen in PCF cells, ruling out a mapping artefact due to the high conservation of the telomere repeats. Taken together, these observations demonstrate the presence of telomere-initiated DNA replication activity in BSF cells that is absent in PCF cells. Given the lack of DNA replication initiation from upstream of any *VSG* BES, we suggest that such telomeric DNA replication only extends significantly beyond the telomere in the single actively transcribed *VSG* BES (here, BES1).

### 177 bp repeats may be widely conserved DNA replication origins in *T. brucei*

To date, genome-wide mapping of DNA replication has not included intermediate- and mini-chromosomes due to lack of assembly[10] in the strain for which MFA-seq data is available[21]. Eight contigs were identified here that contain the 177 bp repeats characteristic of these chromosomes (Fig. 5)[10,44]. Unfortunately, none of these contigs represent

telomere-to-telomere assemblies, and so it cannot be said if they correspond to eight distinct chromosomes or are parts of larger chromosomes. The four largest contigs contain 50 bp repeats and two also contain 70 bp repeats, suggesting they harbour *VSG* BESs (Fig. 5A, B). Three contigs contained telomeric repeats that were not flanked by *VSG*s. All eight contigs are assemblies that incorporate multiple previously unassigned unitigs (Fig. 5C). To explore these chromosomes further, we examined the 177 bp repeats and the gene content.

Sequence analysis by Tandem Repeats Finder[45] indicated that the 177 bp regions in the submegabase chromosomes appear to have two elements: the 177 bp motif itself and a further, shorter repeated region composed of a 59 bp motif that appears telomere-proximal (Supplementary Fig. 9A). Throughout the 177 bp repetitive regions, sequence identity was exceptionally high (>96% across the majority, and >98% in the core). Strikingly, the 177 bp and/or 59 bp repeats could also be identified as components of 11 megabase chromosome centromeres revealed by Nanopore sequencing (Supplementary Fig. 9B, C), with their presence there perhaps previously overlooked due to incomplete assembly. To analyse the gene content of the sub-megabase chromosomes, we used putative protein sequences from Companion genome annotation to identify ortholog groups and paralogs using OrthoMCL. Amongst the genes identified were several *ESAG*s, consistent with the predicted presence of *VSG* BESs (Fig. 5D). In addition, retrotransposon hotspot (RHS) genes were abundant, perhaps indicating subtelomere-like sequences[46,47]. However, the largest group of genes were a diverse selection of hypotheticals (Fig. 5D), the majority of which (24 out of 27 ortholog groups) appear to be specific to *Trypanosoma* species and suggest a wider range of encoded activities. By mapping RNA-seq data from wild-type Lister 427 BSF cells[34], it became apparent that many of the genes not localised to the *VSG* BESs were transcribed (Fig. 5B), which is in striking contrast to the lack of detectable transcripts arising from the megabase chromosome subtelomeres (Supplementary Fig. 10). These data suggest that the Nanopore assembly has revealed a potentially novel class of sub-megabase chromosomes, which are related to mini- and intermediate chromosomes in housing *VSG* BESs and 177 bp repeats but are distinct by virtue of the presence of transcription activity that is not limited to the *VSG* BESs.

As these contigs provide assemblies of sub-megabase chromosomes that are related to the abundant mini- and intermediate-chromosomes, we next tested if they could provide insight into how they are replicated. Though DNA replication bubbles have been detected in isolated mini-chromosomes by electron microscopy[44], they have not been localised. MFA-seq mapping to each contig (Fig. 6A) revealed enrichment of S/G2M reads that was consistent between BSF and PCF cells and appeared to peak across the 177 bp repeats, a localisation that was more clearly seen in metaplots of all the contigs (Fig. 6B). These data suggest that the 177 bp repeats act as origins of replication in these chromosome contigs and, most likely, also in smaller mini-chromosomes and intermediate-chromosomes. The use of 177 bp repeats as origins may also explain DNA replication initiation in at least some megabase chromosome centromeres.

## Discussion

The genome of *T. brucei* is extremely well assembled, having been initially generated through a combination of shotgun sequencing and chromosome-targeted cloning and sequencing[1], and then improved further by a combination of long-read PacBio sequencing allied to Hi-C chromatin capture[9,21,33]. In this context, it is perhaps surprising that Nanopore long-read sequencing can add yet further information, here exemplified by increased understanding of DNA replication dynamics. By assembling contigs that represent much of the megabase chromosomes and some sub-megabase chromosomes, we reveal pronounced compartmentalisation of DNA replication and genome stability (Fig. 7): the transcribed core of the megabase chromosomes is stably maintained by multiple replication origins, many of which share

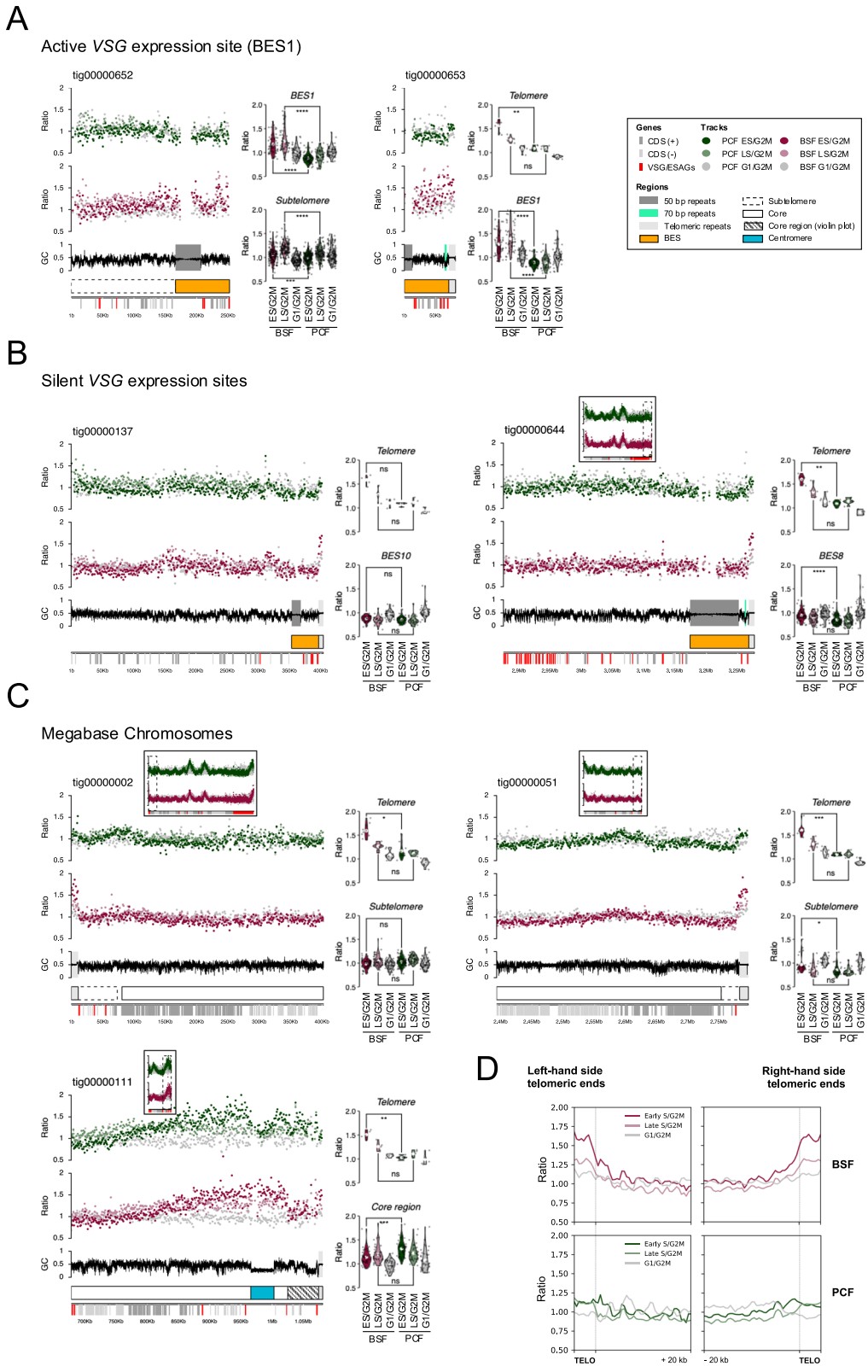

**Fig. 4 | Telomere-directed initiation of life cycle stage-specific DNA replication in the single actively transcribed VSG expression site. A** MFA-seq mapping to two contigs (tigs) that encompass the actively transcribed bloodstream *VSG* expression site, BES1. In tig652 BES1 sequence is truncated at the 70 bp repeats but extends to the subtelomeric region upstream of the 50 bp repeats; in tig653 BES1 sequence extends from the 50 bp repeats to the telomere repeats. MFA-seq is shown for BSF and PCF cells as described in Fig. 2. Violin plots show early S/G2M, late S/G2M and G1/G2M read depth ratios in every 1 kb bin of the subtelomeric region (if present), across the BES, and within the telomere tract (if present).

**B** MFA-seq mapping and violin plots (as above) in contigs that encompass subtelomere, BES and telomere of two silent *VSG* BESs: BES8 and BES10. **C** MFA-seq and violin plots (as above) in contigs that encompass the subtelomere, extending to the telomere tract, of two chromosome ends that do not harbour a *VSG* BES.
**D** Metaplots of MFA-seq signal from BSF and PCF cells mapped across telomere (TTAGGG) repeats and upstream 20 kb of sequence of all telomere-containing Nanopore contigs; the plots are split into contigs where the telomeric sequence is on the 'right' or 'left' end of the contig. Source data are provided as a Source Data file.

177 bp repeats where DNA replication initiates in the sub-megabase chromosomes; in contrast, the unstable, transcriptionally silent megabase chromosome subtelomeres are largely devoid of origins that can be detected by MFA-seq, and the active telomere-proximal *VSG* transcription site that is the target for *VSG* gene rearrangement for immune evasion displays unusual telomere-directed replication early in S phase. Thus, the complex structure of the *T. brucei* genome reflects partitioning into different modes of DNA replication that balance stability and instability (Fig. 7). We propose that this partitioning is needed to allow diversification of the expressed and archive *VSG*s in the unstable genome compartment during immune evasion.

Changes in predicted chromosome structural organisation between the Nanopore Lister 427 assembly here and the Müller PacBio-Hi-C genome[9] may reflect limitations of assembly by either approach[48], or might reveal genuine differences between genome organisation in the stocks of strain between labs, which have been grown independently for decades. Since much of the differences are found in the subtelomeres (Fig. 1), which are known to underlie differences in chromosome size between *T. brucei* strains[8], the latter explanation may be favoured. Nanopore assembly allowed the core and subtelomere compartments of individual chromosomes to be linked in a single contig, which then allowed us to map and compare DNA replication dynamics in the distinct genome compartments (Fig. 2). In contrast to the ready detection of multiple MFA-seq peaks (origins) in each chromosome core[14,19], the only detectable peaks in any chromosome subtelomere coincided with the centromeres. Centromeres are proximal to early replicating origins in yeast[49–51] and are sites of the most prominent MFA-seq peaks in the *T. brucei* chromosome cores[52–54], consistent with early replication. All non-centromeric origins mapped to date in *T. brucei* localise to the ends of multigene transcription units, where transcription initiates or terminates, suggesting an association with the transcription machinery[14]. If so, any such association cannot dictate origin function in the non-transcribed subtelomeres, suggesting that centromeric origin designation may differ from other origins in ways that are not yet clear. In addition, it seems likely that centromeric origins can overcome features, such as heterochromatin, that suppress transcription in the subtelomeres, and perhaps akin to what is seen in *S. pombe*[55]. In *Leishmania*, only a single MFA-seq peak is detectable during S-phase in each chromosome[53,56], and each of these origins also appear to coincide with a centromere[54]. Thus, kinetoplastids may provide evidence of proposed ancestral overlap between origins and centromeres[57], with *T. brucei*, but not *Leishmania*, having evolved further origins that have been separated from centromeres in ways not yet explored.

By mapping DNA replication dynamics across the entirety of the *T. brucei* megabase chromosomes, we reveal differing origin density in the core and subtelomeric compartments of the genome, which is associated with differing levels of instability (Fig. 3). Whereas the origin-rich core is largely stable during growth, the origin-poor subtelomeres are more unstable. We have shown previously that mutation of *T. brucei* BRCA2 leads to loss of some *VSG*s from the genome[41,42]. Improved assembly of the subtelomeres now reveals the scale of this gene loss and shows that the extent of such instability is even greater in RAD51 mutants. Further work will be needed to determine if there is a causal link between origin density, homologous recombination and genome instability in the subtelomeres relative to the core, but there are intriguing parallels in such compartmentalisation of mutation level and origin localisation with that recently described in archaea[39]. Indeed, theoretical[58–60] and experimental analyses[61] have shown that regions of eukaryotic chromosomes devoid of, or denuded in origins display increased mutagenesis and instability. Moreover, DNA damage response factors are needed to support replication of yeast artificial chromosomes lacking origins[62]. Nonetheless, it may be that compartmentalisation of the *T. brucei* genome into a highly transcribed core of mainly 'housekeeping' genes and a subtelomere compartment that mainly houses virulence genes (*VSG*s), means that gene loss in the latter is more readily tolerated than the former.

More immediate parallels between DNA replication dynamics and (in)stability in the *T. brucei* genome may be found in two related trypanosomatids, *Leishmania* and *T. cruzi*. *Leishmania* have notably unstable genomes, due to fluctuating levels of aneuploidy[63–65] and genome-wide gene copy number variation[66]. Though recent data have revealed the phenotypic consequences of aneuploidy and the rates and levels at which it arises[67–72], the range of mechanisms that lead to such genome instability is unclear. Copy number variation through episome formation is driven by homologous recombination[73–76] and recent work has revealed links between recombination and the programming of *Leishmania* DNA replication[56,77]. Moreover, MFA-seq mapping indicates just a single clear S-phase initiation locus in each chromosome, which is unlikely to be able to support efficient DNA replication of all chromosomes[53,56]. Testing whether or not genome-wide origin paucity might dictate the extreme instability of the *Leishmania* genome appears worthwhile[12]. The *T. brucei VSG*-rich subtelomeres may have more obvious parallels with the multigene gene families that provide a 'disruptive' feature of the *T. cruzi* genome[22] and, indeed, it has been proposed that origin localisation may relate to the well-known propensity of the families to undergo recombination and rearrangement[78,79]. However, there are key differences between organisation of the putative disruptive gene family-rich elements of the genome in the two parasites. First, unlike in *T. brucei*, origins appear to localise to at least some of the *T. cruzi* multigene families[78]. Second, in *T. brucei*, silent *VSG*s are predominantly localised to the subtelomeres, meaning this genome compartment is geographically isolated from the transcribed core of chromosomes, whereas in *T. cruzi* the disruptive compartment is found within and across the chromosomes and, hence, is intermingled with the core[80]. Nonetheless, Hi-C provides clear evidence for interaction boundaries between the core and subtelomeres of the *T. brucei* chromosomes[9], and the disruptive and core compartments of the *T. cruzi* genome[80]. Despite this, the separation of core and disruptive elements of the *T. brucei* and *T. cruzi* genomes appears not to be driven by detectable sequence features, despite differences in chromatin[81,82]. These findings have two implications. First, how distinct chromatin organisation is determined in the two compartments is unclear in both parasites. Second, the distinction in chromosome organisation in the two parasites may reflect differing features of the usage of the gene families, such as the roles of surface antigens during immune evasion and mechanisms or frequency of recombination.

It is often stated that the sub-megabase chromosomes of *T. brucei* evolved to expand the archive of silent *VSG*s and *VSG* BESs needed for immune evasion[4,10,44]. In addition, the function and evolution of the 177 bp repeats has not been determined[10]. Here, long-read Nanopore sequencing allowed assembly of several 177 bp repeat-containing chromosomes that expand our understanding of the abundant sub-megabase chromosomes of *T. brucei* (Fig. 5). These assemblies incorporate several previously unassigned unitigs, and around half of the chromosomes contain *VSG*s and telomeric *VSG BES*s, as previously described. However, all the chromosomes contain a number of further, transcribed genes that mainly encode a range of hypothetical proteins, and in several cases these gene-rich components of the chromosomes are adjacent to the telomeres. Thus, all these chromosomes do not merely house *VSG*s or *ESAG*s, perhaps suggesting that mini- and intermediate-chromosomes are part of wider continuum of sub-megabase chromosomes, some of which provide functions beyond immune evasion. Previous analysis has, in fact, suggested the presence of such gene-rich elements: detailed restriction mapping of many mini-chromosomes revealed stretches of non-repetitive DNA in several of them[10], including evidence of a large subtelomeric stretch[10,83], but without determining the sequence composition. Cloning and

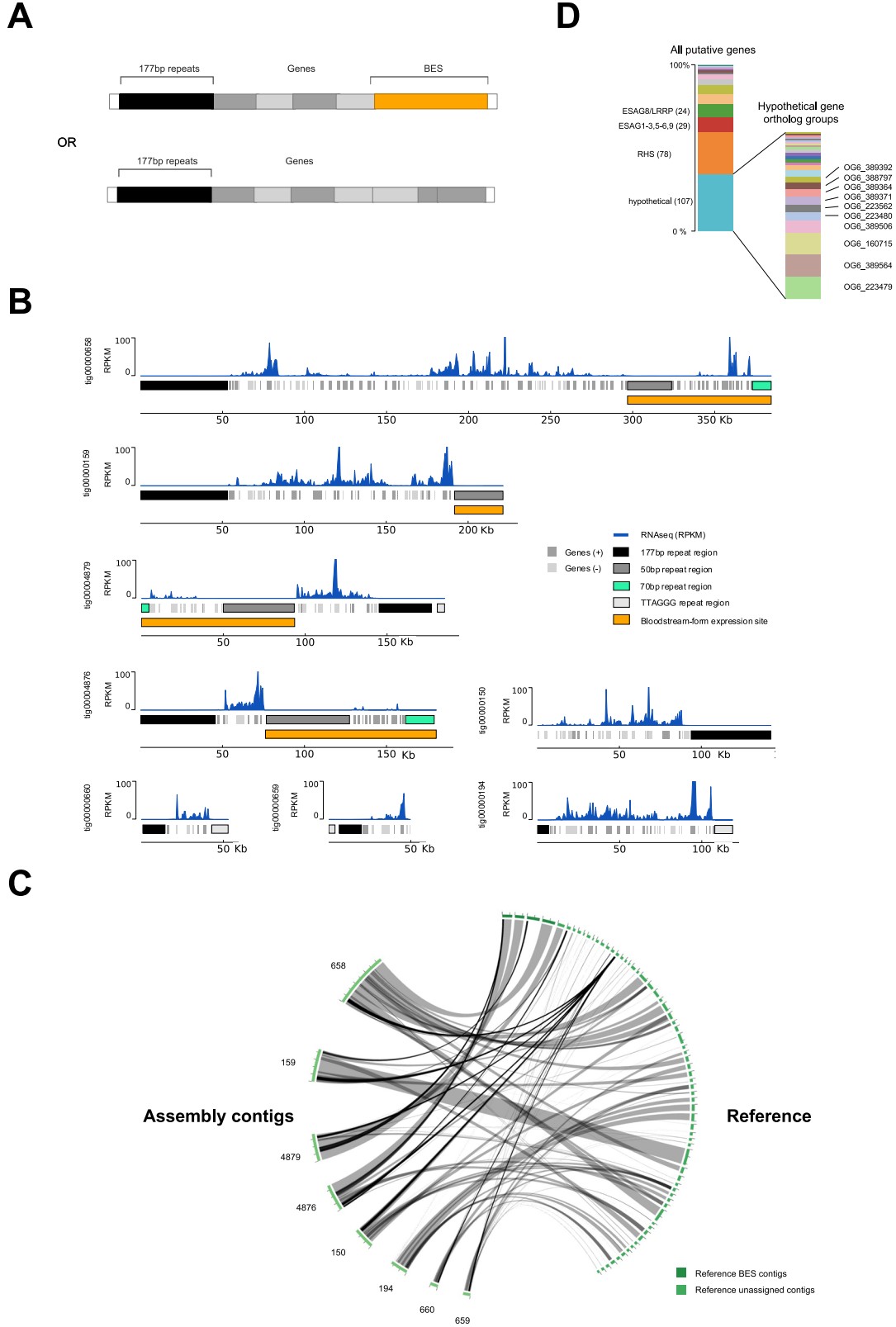

**Fig. 5 | Assembly of 177 bp-containing sub-megabase *T. brucei* chromosomes that harbour diverse transcribed genes. A** A diagrammatic representation of two distinct forms of Nanopore contigs, one type housing a telomeric *VSG* BES (orange) and each containing 177 bp repeats characteristic of sub-megabase chromosomes of *T. brucei*. **B** RNA-seq mapping to eight sub-megabase chromosome Nanopore contigs (tigs). **C** Circos plots showing correspondence between the sub-megabase chromosome Nanopore contigs and many previously unassigned unitigs; grey ribbons represent simple overlaps, and black ribbons represent multiple overlaps within a region. **D** Gene content of the novel, smaller chromosomes: only the most abundant predicted gene types are detailed (total numbers are shown), amongst which the range of orthology groups (OG6) predicted amongst hypothetical genes by OrthoMCL is illustrated. Elements of panel A were created in BioRender. McCulloch, R. (2025) https://BioRender.com/i58h117.

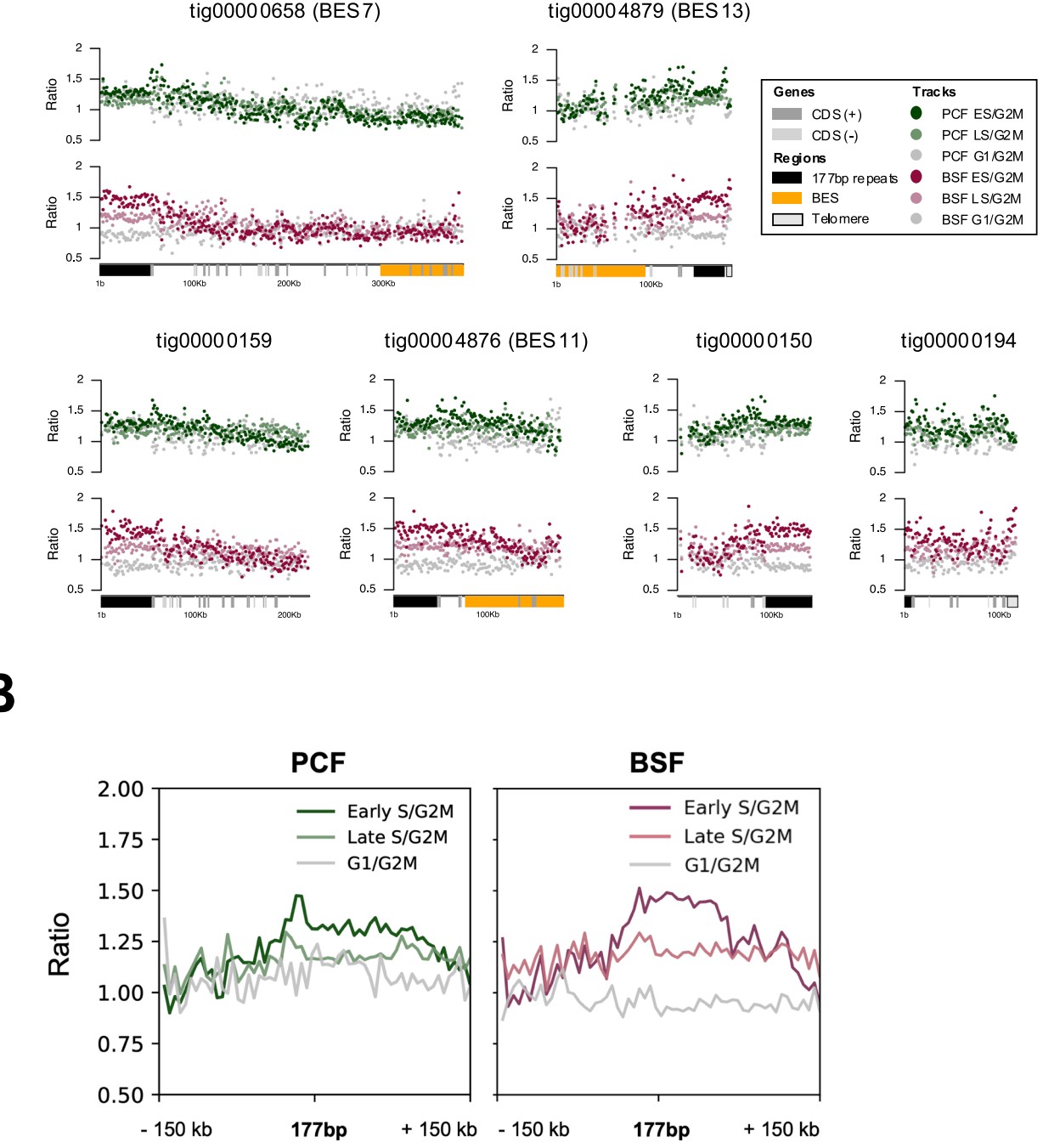

**Fig. 6 | The 177 bp repeats are widely conserved origins of DNA replication in *T. brucei.* A** MFA-seq mapping is shown across each of the eight Nanopore contigs (tigs) of sub-megabase chromosomes; representation of genes, *VSG* BES and repeats is as shown in Fig. 5, and early S/G2M, late S/G2M and G1/G2M read depth ratios in BSF and PCF cells is as shown in Fig. 2. **B** Metaplots of MFA-seq signal from BSF and PCF cells mapped across the submegabase contigs, centring the data on the 177 bp repeats and showing 150 kb of upstream and downstream sequence. Source data are provided as a Source Data file.

sequencing of a bacterial artificial chromosome allowed the characterisation of *VSG* BES2 (containing *VSG* 427-9/VO2) and upstream, subtelomeric sequence[84]. BES2 was not detected in our Nanopore assembly, but it has been shown to reside on an intermediate chromosome, and the presence of RHS genes upstream of the 50 bp repeats appears consistent with these genes being detected in at least

some of the sub-megabase chromosomes we describe here. These data lend weight to the suggestion that the small chromosomes of *T. brucei* arose from parts of the megabase chromosomes[84]. A further argument for such evolution is the demonstration here that 177 bp repeats, thus far considered characteristic of mini- and intermediate-chromosomes[10], are components of at least some centromeres in the

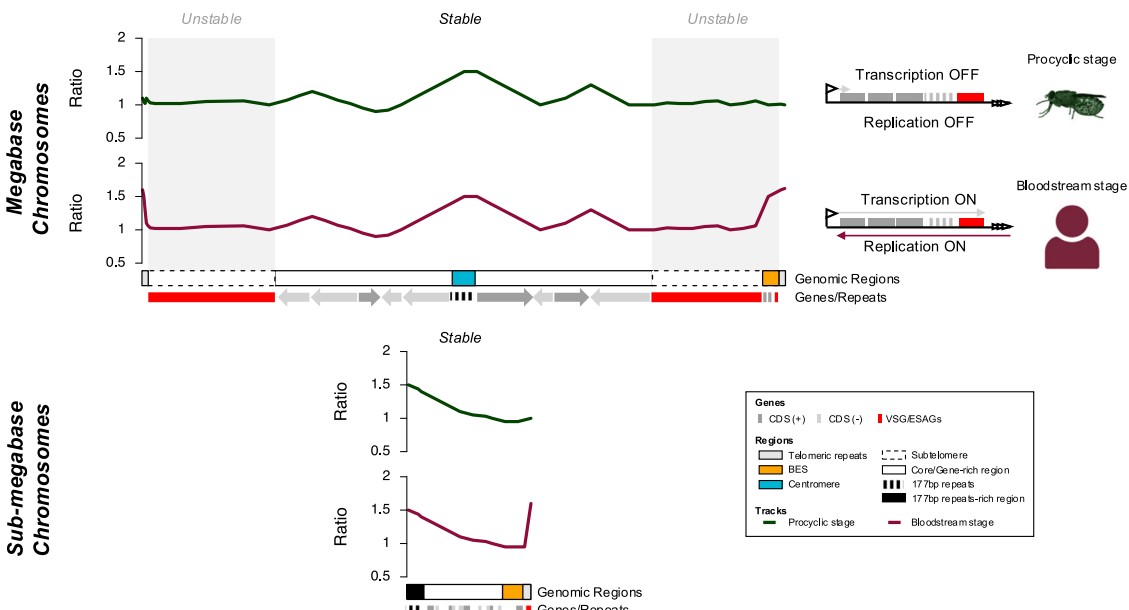

**Fig. 7 | DNA replication programming in *T. brucei* dictates compartmentalised genome stability.** A summary of DNA replication functions that denote stability and instability across the *T. brucei* genome. The transcribed cores of the megabase chromosomes are replicated from multiple origins, the earliest acting of which coincide with centromeres. Many of the centromeres contain 177 bp repeats that are found in all sub-megabase chromosomes, where the repeats act as DNA replication origins. The subtelomeres of the megabase chromosomes are largely devoid of origins and are unstable relative to the cores. *VSG* bloodstream expression sites (BESs) are found adjacent to the telomeres of megabase and sub-megabase chromosomes. Only the single actively transcribed *VSG* BES displays early S-phase DNA replication, which is derived from the telomere and likely results in head-on collisions with monoallelic BES transcription. Early replication of the single *VSG* BES transcribed in mammal-infective parasites does not occur in parasites within the tsetse fly vector, where transcription of all *VSG* BESs is silenced. Elements of the figure (the fly and the human) were created in BioRender. McCulloch, R. (2025) https://BioRender.com/w43b159.

megabase chromosomes. Furthermore, we provide evidence that the 177 bp repeats act as DNA replication origins in the newly assembled sub-megabase chromosomes (Fig. 6). It seems likely, therefore, that such origin activity explains how all these abundant small chromosomes are copied (Fig. 7) and why they show considerable mitotic stability[11].

The suggestion that the 177 bp repeats are a previously undetected, widely conserved DNA replication origin has implications both for *T. brucei* DNA replication programming and the mechanism of DNA replication initiation. DNA replication mapping has shown the megabase centromeres to be either the earliest replicating origins, or the most efficiently used origins[14,19]. Such centromeric origin activity may derive from the 177 bp repeat element of these larger structures, explaining why the repeats are found in all the abundant small chromosomes and suggesting they too may be early replicating. Whether the 177 bp repeats might also provide kinetochore-like activity needed for the segregation of duplicated chromosomes is unknown[85]. In addition, further work will be needed to determine if sequences derived from or related to the 177 bp repeat might also guide replication initiation in centromeres that appear to lack the repeat, and in non-centromeric origins, which appear to replicate later in S-phase. For instance, it will be interesting to test if the 177 bp repeats provide a conserved sequence feature for ORC binding[52,86], thereby dictating origin activity. Irrespective, the conserved 177 bp repeats appear to be the most abundant origins in the *T. brucei* genome. So far, the only other eukaryotes known to possess sequence-conserved origins are yeasts related to *Saccharomyces cerevisiae*[57].

The availability of Nanopore contigs that span subtelomeres, 50 bp repeats, *VSG* BES and the telomere repeats provided new understanding of *VSG* BES replication (Fig. 4). Nanopore assembly allowed a fuller description of the 50 bp repeats[29] upstream of the *VSG* BES, showing them to be highly conserved. What this conservation

might mean in terms of function is unclear, but we could detect no evidence for DNA replication initiation at or around these elements, indicating they do not act as origins for *VSG* BES replication. In fact, in several cases we could map DNA replication from an upstream subtelomere sequence that did not extend into a downstream, silent *VSG* BES. Perhaps this suggests that the proposed 'boundary' function of the 50 bp repeats[29] is to limit early S-phase replication of the active *VSG* BES from encroaching into the upstream subtelomere/core. We also note that the 50 bp repeats are unique amongst the repeats we have examined in the *T. brucei* genome in not being a locus of RNA-DNA hybrid accumulation (Fig. S3B), for reasons that are unclear. Rather than arising from the 50 bp repeats or from the subtelomeres, our new data indicates that BSF-specific DNA replication, which is only seen in the single active *VSG* BES, arises from the *T. brucei* telomere (Fig. 7). The evidence for this is two-fold: first, MFA-seq signal across the *VSG* BES diminishes with distance from the telomere tract and, second, we can detect MFA-seq signal at all telomeres in BSF cells. Two explanations might then be considered for how the active *VSG* BES alone is replicated. DNA replication may initiate from all telomeres in BSF cells but is curtailed from extending into inactive *VSG* BESs, perhaps due to more repressive chromatin or the novel base J[87]. Alternatively, it is possible that only the active *VSG* BES-associated telomere directs replication and the MFA-seq signal we see at all telomeres is simply cross-mapping. In either scenario, whether or not ORC is responsible for initiating the DNA replication is unclear; however, ORC has been described to interact with telomeres in other eukaryotes[88,89], and loss of *T. brucei* ORC impairs telomere integrity and leads to altered VSG expression[90]. If and how such ORC binding and/or DNA replication activity might be limited to *T. brucei* BSF cells and curtailed in PCF cells also remains to be determined. Nonetheless, whatever the mechanics of the reaction prove to be, telomere-directed DNA replication of the active *VSG* BES seems likely to cause DNA polymerase (and perhaps a complete replisome) to be in head-on conflict with RNA Polymerase I

transcription, providing an attractive model for the generation of focused damage and recombination that drives *VSG* switching and immune evasion[91,92].

## Methods

### Parasite culture, DNA extraction and sequencing

Monomorphic bloodstream-form (BSF) *Trypanosoma brucei brucei* cells (strain Lister 427) were propagated at $37°$ C and 5% $CO_2$ in HMI-9 medium with 10% foetal calf serum. Cell growth was analysed using a haemocytometer at 24 h intervals. Every two days the cells were passaged, reducing cell density to $1 \times10^4$ cells mL$^{-1}$. *BRCA2-/-* (Tb427_010006100)[41,42] and *RAD51-/-* (Tb427_110089600) cells were generated previously[40].

For genomic DNA extraction, 200 mL of culture per sample was harvested at approx. density of $1 \times10^6$ cells mL$^{-1}$. The samples were centrifuged for 10 min at room temperature at 1000 x g, the supernatant was then discarded, and the cell pellet resuspended in 400 µL of 1X phosphate buffered saline (PBS) followed by centrifugation for 3 minutes at room temperature at 1000 x g. The supernatant was discarded and the cell pellet stored temporarily at -20° C. For short-read sequencing DNA extraction was performed using a Qiagen DNeasy kit as per manufacturer's instructions (Animal Blood or Cells Spin Column protocol). For initial (P0) and late (P23) passage samples, DNA library was prepared using a Qiagen QiaSeq FX DNA library kit and an Illumina NextSeq 500 was used to perform paired-end whole genome sequencing (2x75bp). Subclones of later passage (P23) were sequenced using DNBSEQ (2x100bp).

For ONT sequencing, DNA extraction was carried out using the Qiagen MagAttract HMW DNA kit as per manufacturer's instructions. During library preparation, ONT's Ligation Sequencing Kit (SQK-LSK109) and Rapid Barcoding Kit (SQK-RBK004) were used as per manufacturer's instructions, and sequencing was performed on a MinION 1B device using R9.4.1 MinION and Flongle flow cells. DNA quality and quantity was assessed using NanoDrop 2000, Qubit 3.0 Fluorometer (BR dsDNA kit) and BioAnalyzer 2100 (High Sensitivity DNA kit).

### Genome assembly and QC

Basecalling was performed using guppy (version 3.3.3 for Linux CPU) using the high accuracy settings. Quality control and basic sequencing metrics on basecalled data were produced using NanoPlot (version 1.30.0)[93]. Long-read genome assembly was performed using canu[30] with default settings and predicted genome size of 35 Mb. Four iterations of polishing using pilon (version 1.23)[31] were performed using paired-end Illumina data (2x75bp), and the resulting genome assembly was used in all subsequent analyses. Quality control and assessment was done using BUSCO (v. 3.5.2, database – eukaryota_odb10)[32] and QUAST (v. 5.0.2.)[94].

In order to characterise mapping of very long reads across the genome assembly, the ONT sequence data that was used for genome assembly was mapped using minimap2 (minimap2 -ax map-ont -t 32 basecalled.fastq > aligned.sam)[95], and further converted to bam, sorted and indexed using samtools[96]; a custom script using samtools and awk were used to extract reads >50 kb in length, as well as the mapping quality of the reads. Matplotlib (v. 3.5.3) was used to plot these data. For read depth coverage of >50 kb reads across the genome, samtools depth was used on the filtered bam file, and this data was then analysed using pandas (v. 2.0.3) to extract statistics regarding read depth coverage across the genome.

To assess base quality at reads spanning 70 bp and telomeric repeats, reads >10 kb were extracted as described above for >50 kb reads, except only for regions covering the repetitive elements and 20 kb flanking sequence on both sides (for visualisation purposes). Base quality and read depth coverage of these reads was evaluated using the python package pysam (v. 0.22.1) in 100 bp bins and the median values, along with the interquartile range, plotted for each bin using matplotlib (v. 3.5.3). To assess base quality of reads spanning the telomeric repeats relative to the 70 bp repeats, the base quality data was plotted as boxplots, and the difference tested using a Mann-Whitney U test using the python scipy package (v.1.10.1).

### Genome annotation and comparison

Companion[97] was used for general genome annotation. *Variant surface glycoprotein (VSG)* genes and pseudogenes in the assembly were identified by extracting *VSG* sequences from the TriTrypDB reference genome for *Trypanosoma brucei brucei* Lister 427 2018 (build version 47), creating a Basic Local Alignment Search Tool (BLAST) database of *VSG* sequences using makeblastdb, and using that to identify *VSG* sequences in the new assembly using blastn; this approach also identifies putative *ESAG6* and *ESAG7* genes in some cases. To analyse gene content of the novel, smaller chromosomes, protein sequences from Companion annotation were used as input for the OrthoMCL (OG6_r20) pipeline on VEuPathDB's Galaxy (https://veupathdbprod.globusgenomics.org)[98]; the gene product description for the matching *T. brucei* gene on TriTrypDB was used to describe putative genes on the smaller chromosomes.

In order to characterise any novel putative genes in the Nanopore assembly, we performed reciprocal BLASTp searches between the protein sequences from the TriTrypDB version of the Muller genome and the new assembly. Any protein sequences found not to have hits in the Muller genome were further analysed in VEuPathDB's OrthoMCL pipeline that maps protein sequences to known OrthoMCL groups using DIAMONDp (https://orthomcl.org/orthomcl/app/workspace/map-proteins/new).

Synteny relative to the published Muller reference genome was analysed by running reciprocal alignments with minimap2 (command: minimap2 -x asm5 ref.fa assembly.fa > aln.paf)[95] and visualised using Circos[99]. Reciprocal assembly comparison was used in order to identify *VSG* BESs, confirm contig identity, assess potential rearrangements, and gap bridging.

### Repeat identification and characterisation

Initial repeat identification in the genome was performed using Tandem Repeats Finder (TRF, version 4.09)[45] using the recommended settings. The localisation of repeats reported by TRF was then investigated in the assembly-vs-assembly alignment in order to identify the genomic context of the identified repeats. Motif sequences identified by TRF were subsequently used to further investigate underlying repetitive region structure using FIMO of the MEME suite of tools[100], identifying the individual occurrences of the motifs in the region and genome more broadly; filtering based on maximum p-value ($<10^{-9}$) was applied to FIMO searches. In order to locate centromere-associated repeats, makeblastdb was used to create a blastn database containing flanking regions of reference scaffold gaps in putative centromeric loci in the Muller genome. The database was used to query the new assembly with blastn; individual matches were manually checked to discard genic sequences, compared to TRF output, and the final coordinates, refined using TRF, were used in subsequent analysis.

### Repeat sequence composition analysis

Overall AT content of repetitive regions was determined using seqtk comp, by subtracting the number of G and C nucleotides from the total length of the region. Genome-wide GC content, as well as AT and GC skews, were calculated using nuc from the BEDtools suite of tools[101] with 50 bp bin size; bedtools makewindows was used to create the genome-wide bedfile containing all non-overlapping 50 bp intervals. AT skew was calculated as follows: AT skew = $(A − T)/ (A + T)$, whereas GC skew was calculated as follows: GC skew = $(G − C) / (G + C)$, where A, T, G and C represent the number of occurrences of the respective nucleotide in a given window. To assess repeat conservation of 177 bp

repeat regions, StainedGlass[102] was used; briefly, repetitive regions were split into 500 bp bins, an intra-region all-vs-all alignment was performed using minimap2, and % identity across the repetitive regions was calculated and visualised using StainedGlass.

## Short read data processing

Illumina reads were trimmed and filtered using trim galore in paired-end mode with fastQC enabled (v 0.6.10), aligned to TriTrypDB reference genome *T. brucei brucei* Lister 427 release 46 using bwa mem in paired-end mode (v. 0.7.17-r1188)[103], formatted to bam, sorted and indexed using samtools (v. 1.19.2)[96]. DeepTools bamCoverage and bamCompare[104] were used to analyse read depth coverage (RDC) and changes in RDC, respectively (normalisation using read count for bamCompare and RPKM for bamCoverage; minimum mapQ 1, bin size 50 bp). DRIPseq data [34] was processed as above, except the normalisation method used for bamCompare was SES. RNAseq data [34] was trimmed using trim galore as above, aligned using hisat2 (v.2.2.1) (settings: --no-spliced-alignment)[105], with further processing to sorted bam file as above, with additional samtools filtering retaining only reads that map once in the genome (samtools view -Sbu -d NH:1 file.sam > file.bam). DeepTools bamCoverage was used to assess expression across the genome, with RPKM used for normalisation.

## MFAseq data processing

The data used in the MFA-seq analysis is available in ENA project PRJEB11437[19], and was processed as described[56], with a small number of adaptations. Except for the MFA-seq ratio calculation (below), all data processing was performed on the Globus genomics platform via VEuPathDB. The data was first trimmed using Trimmomatic[106] (standard settings, paired-ended, minimum quality of 20), then aligned to the long read Nanopore assembly or the TriTrypDB release 46 *T. brucei brucei* Lister 427 2018 genome using Bowtie2[107] (paired-ended, local, very sensitive). The aligned data was then processed with bamCoverage (deepTools)[104] with a bin size of 1 kb (no scaling/normalising method, paired-ended extension, ignore duplicates, centre regions with respect to fragment length, and a minimum MapQ of 1). The output files were then screened for bins of sizes different from 1 kb, as well as bins with less than 100 reads per bin. Bedtools-intersect (bedtools)[101] was used to remove these bins from all data sets. The remaining data files were then exported from Globus genomics and MFA-seq ratios calculated using the mfaseq_bed_py3.py script designed by Dr Kathryn Crouch (https://github.com/kathryncrouch/misc) where the G1, Early S (ES) and Late S (LS) data sets were normalised to the G2M data set.

For visual representation, KaryoploteR[108] was used via RStudio to represent the data across genomic regions, while deepTools computeMatrix and plotHeatmap were used to generate metaplots. For ease of comparison of the MFA-seq data across certain chromosome features (e.g. telomeres, subtelomeres, BES), the data within the coordinates of these features was represented as violin plots. The data was processed and plotted in RStudio using ggplot2. Statistical analysis—non-parametric Kruskal-Wallis test followed by multiple pairwise comparison (all groups considered) analysis by Dunn's test with Bonferroni correction (ns, not significant; *, $p$ value < 0.05; **, $p$ value < 0.01; ***, $p$ value < 0.001; ****, $p$ value < 0.0001)—was performed in RStudio using the packages rstatix, ggsignif and ggpubr.

## Reporting summary

Further information on research design is available in the Nature Portfolio Reporting Summary linked to this article.

## Data availability

Nanopore and Illumina reads have been deposited to the NCBI Sequence Read Archive (SRA) under project number PRJNA962304.

The assembled genome is available at the EMBO-EBI European nucleotide archive (ENA), accession number PRJEB75536. MFAseq sequencing data is available on the ENA under PRJEB11437. DRIP-seq and RNA-seq datasets are available under PRJEB21868. Source data are provided with this paper.

## Code availability

The python script used to process MFAseq data is available at Zenodo at https://doi.org/10.5281/zenodo.14224855[109].

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

## Acknowledgements

We thank J. Galbraith and D. McGuinness (Glasgow Polyomics, University of Glasgow) for help with sequencing, and all members of the McCulloch lab for discussions. This work was supported by the Wellcome Trust (224501/Z/21/Z to RM, 218648/Z/19/Z to EMB), a Wellcome Trust Institutional Strategic Support Fund award held at the University of Glasgow (204820/Z/16/Z to RM, EMB, GH and KC), and the BBSRC (BB/N016165/1, BB/R017166/1 to RM, and BB/W001101/1 to RM and CAM). The Wellcome Centre for Integrative Parasitology was supported by core funding from the Wellcome Trust [104111].

## Author contributions

Conceived the study: M.K., C.M., G.H., R.M. Designed and conducted the research: M.K., C.M., E.B., C.L. Analysed the data: M.K., C.M., E.B., D.B., K.C., R.M. Wrote the initial draft: M.K., C.M., R.M. Edited and approved paper: M.K., C.M., E.B., D.B., K.C., R.M. Funding: C.M., E.B., R.M.

## Competing interests

The authors declare no competing interests.
