## [Transparent Peer Review file · Nature Communications]

Nanopore sequencing reveals that DNA replication compartmentalisation dictates genome stability and instability in *Trypanosoma brucei*

Corresponding Author: Professor Richard McCulloch

Version 0:

Reviewer comments:

Reviewer #1

(Remarks to the Author)

This manuscript describes the use of nanopore sequencing technology to dissect the dynamics of DNA replication in different genomic compartments of the *Trypanosoma brucei* genome. The work has revealed new insights' showing the replication of the active VSG expression site is directed from the telomere, that the core and subtelomeric regions display different levels of genomic stability which may relate to the use of subtelomeres for the immune evasion genes. They have demonstrated that the 177bp repeats act as replication origins and furthermore that these are found in the centromeres of the larger chromosomes too. The data support the conclusions drawn. This is a very carefully performed study which adds greatly to the understanding of the trypanosome genome and generates useful testable hypotheses.

Minor point: are the newly discovered hypothetical transcribed genes conserved across kinetoplastids or are they a *T. brucei* specific set of genes.

I recommend acceptance of this manuscript.

Reviewer #2

(Remarks to the Author)

Trypanosoma brucei has a complex genome organisation, with most protein-coding genes located in transcribed chromosome cores while large archives of silent variant surface glycoprotein (VSG) antigen genes and VSG transcription units are located in arrays present in the heterozygous subtelomeric regions. In addition, the *T. brucei* genome contains several repetitive elements spanning many kilobases, including 70 bp repeats, 177 bp repeats, 50 bp repeats, centromeres and telomeres.

Despite significant advances in assembling a complete *T. brucei* genome, a telomere-to-telomere assembly has not been reported, meaning mapping of NGS data to these repetitive/unassembled regions cannot be accurately determined.

Here, Krasilnikova et al., use long-read Nanopore sequencing to assemble many of these repetitive elements and subtelomeric loci. The authors have previously mapped origins of replication as well as DNA replication dynamics using MFA-seq, however this analysis was mostly limited to the transcribed core regions as well as the VSG transcription units. The authors now provide a more complete picture of these replication dynamics by re-mapping previously published MFA-seq data to this new assembly.

As all papers from the McCulloch lab, this manuscript is very well written and contains a beautiful introduction and extensive discussion that nicely put the work in perspective.

Although I have some questions and concerns (see below), I believe the work presented here represents a very important step towards a proper understanding of genome replication in trypanosomes.

My main concerns relate to the accuracy evaluation of the new genome assembly and the claim that subtelomeric regions are more unstable than the chromosome cores.

Major points

1) Genome assembly:

a) Quality check of the assembly

Based on Fig. 2, contig Tig00004860 contains DNA from chr 1, chr 6 and chr 7, contig Tig00000002 contains DNA from the core of chr10 and subtelomeric DNA from both haplotypes 5A and 5B at the same time? Similarly, based on Fig1a the core of chr 4 (tig58) contains subtelomeric DNA from both haplotypes 5A and 5B at the same time.

It is very surprising that DNA sequences from different chromosomes are merged into one contig. If this is real, it would suggest that the original Müller was incorrectly scaffolded. How did the authors evaluate the correctness of the assembled contigs? If it was incorrectly assembled here, could it be that other contigs are also incorrectly assembled?

The authors used Busco to evaluate the quality of the assembly, but I believe that it only assesses the completeness of the housekeeping gene assembly and does not yield information about the overall correctness of the genome assembly.

b) Table S2.

- The authors show that there are an additional 3708 protein-coding genes in their new assembly compared to the 427 2018 assembly. Where/what exactly are these genes?
- For the total length, is the sequence of the two homologous chromosomes counted separately?
- In the Müller genome, Ns were used to indicate gaps in the scaffold. Thus, the statistic about the number of reduced Ns is not very informative.
- I think that the differences in the number of contigs number is only partially informative. For example, in the Müller genome, cores and subtelomeres were separated for easier visualization of data. Also, the new nanopore assembly seems to have a much smaller number of contigs containing 177 bp repeats (8 compared to ~60 in the Müller genome. These are presumably contigs from intermediate or minichromosomes. Why are these contigs missing? In contrast, the new nanopore assembly seems to be much better at predicting the length of repeat regions especially, the 50 bp repeats seem to be much better covered in this assembly.

c) I find Fig 2 difficult to read. The labels are very small and it is not always clear where the boundaries between cores and subtelomeres are. As a consequence, the 'compartmentalization' is not clearly visible to me. Could the authors label the boundaries? Also, I don't find the contig label very useful. The compartmentalization' is easier to see in Fig S5. Perhaps the authors could merge the information from both figures and highlight where the new assembly has provided additional insight.

d) Line 316. The authors write "These data suggest that the Nanopore assembly has revealed a potentially novel class of sub-megabase chromosomes,...". What makes the authors think that this is novel class of chromosomes and not simply intermediate chromosomes? However, it is very interesting that the authors identify actively transcribed regions within these chromosomes. It is important that the authors clarify if they are filtering out multimapping reads or not. If all reads are used, it could be that the reads are from another genomic locus.

e) Line 241 What do the authors mean by 3 BES were duplicated? Do they mean that they are duplicated in the genome or that they have multiple contigs covering parts of the BESs?

2) Instability of subtelomeric regions

Line 234: Instability of subtelomeric regions

I am not convinced that the data really demonstrate higher instability in subtelomeres compared to cores in WT. Perhaps I am missing something, but I only see a decrease in coverage in one clone in Fig 3, nothing in Fig S7. For the RAD51-null mutant there are clear differences. However, a difference in coverage does not really mean that there is lower stability, the subtelomeres are haploid, thus any loss will be much more visible. In addition, any loss in the core region would be likely associated with a severe growth defect and would therefore be selected against.

3) 177 bp repeats as origins of replication.

a) The origins of replication described in the 177bp repeats are intriguing, but the only experimental evidence the authors provide that these act as origins is MFA-seq. It would be nice to see ChIP-seq against an ORC protein (or other similar DNA replication protein) to see if the MFA-seq matches ORC binding in these regions. The authors could map the available data. Also, can the authors really rule out the possibility that the telomeric sequence is acting as an origin of replication?

b) It would be nice to see more information on possible mechanisms of how repetitive regions of the genome can act as of origins of replication in other organisms and in trypanosomes (either in the introduction or in the discussion).

c) Line 264. The authors report a decrease in MFA-seq reads in the subtelomeres. To better see this drop, it would be good to plot the data next to each other.

d) Line 274. Did the authors do any filtering for mapping quality (uniqueness of reads)? I assume that the drop in read

coverage over the 50 bp is caused by the authors using only uniquely mapping reads. If this is the case, how were the authors able to analyse coverage across telomeric repeats?

e) Line 284. I do not understand how the authors can conclude that there is general telomere-initiated DNA replication activity in BSF. Couldn't the signal come from the active BES? Or from early replicating MC? This problem is acknowledged by the authors in the discussion but it should be considered earlier.

f) Line 29. The authors write "To date, genome-wide mapping of DNA replication has not included intermediate- and mini-chromosomes due to lack of assembly in the strain for which MFA-seq data is available". However, Cosentino et al (2021) describe the identification of 60 contigs with 177 bp repeats for the Lister 427 strain. Why did authors not map their MFA-seq data to these contigs?

4) The sequencing data is not accessible at EBI under the accession number PRJEB75456.

Minor points

Line 56: The authors state that the parasite has genome of 60 Mb and cite Berriman et al. However, this reference states that the genome is 26 Mb in size. Are the authors referring to the combined size of both homologous chromosomes?

Line 37-39: This sentence is slightly misleading. The 177bp repeats are present on minichromosomes and intermediate chromosomes, however VSG transcription sites are only located on intermediate chromosomes and not minichromosomes. This is not made clear here, either the sentence should be reworded or a distinction should be made between these chromosome types.

Line 105-107: this sentence did not read so well for me. I would probably rephrase

In Fig S1. there is a formatting error in the number displayed on the bottom left hand side of each chromosome schematic.

Line 599: BESS should be BESs

Line 625: Should this be "Telomere-directed initiation" instead of "Telomere-directed the initiation"?

All instances of "HiC" should be "Hi-C"

Reviewed by James Budzak Nicolai Siegel

Reviewer #3

(Remarks to the Author)

This manuscript leverages nanopore long reads to improve the genome assembly for *Trypanosoma brucei*. The authors then demonstrate the utility of the improved assembly by re-mapping previously acquired genomic datasets to gain new insights into trypanosome biology. The *T. brucei* genome is particularly complex (and distinct from most other eukaryotes), with long poly-cistronic transcription units, complex repetitive sequences and many smaller (poorly characterised) chromosomes. One crucial biological aspect of *T. brucei* is the ability to switch the variant surface glycoproteins (VSGs) to avoid the host immune response. There remains much to be learnt about the mechanisms that participate in the switching of the expressed VSGs, and this manuscript makes some valuable contributions in this area.

Overall, the manuscript feels a little like a somewhat uncomfortable combination of two stories. First, the improved genome assembly that whilst it could be seen as incremental has the potential to be a value to the research community. Second, a new characterisation of replication dynamics using existing datasets now mapped to the new assembly. Ultimately there is value in this combination, since it allows the authors to demonstrate the importance of improved genome assemblies for gaining greater insight into fundamental biological processes.

However, we feel that there could be some improvement to the structure of manuscript to help make it more accessible to a wider audience. The submitted version switches back and forth between describing new aspects of the assembly and new insights from re-mapped data. It would probably be clearer and more accessible, if the manuscript cleanly described the new assembly (e.g. the main chromosomes and novel genes identified on the smaller chromosomes) in an initial section, and then moved onto the more functional analyses that rely upon the remapping of existing datasets. Subject to this comment and those below, we feel that this is an interesting manuscript deserving of publication.

Major comments:

1. The manuscript would benefit from increased clarity of language and some simplification to figures. This would help make the manuscript more accessible to a wider audience. We recognise that the *T. brucei* genome is complex in nature and function, differing from most model species. We suggest carefully introducing the key differences between the *T. brucei* genome and common models avoiding wherever possible complex terminology, e.g. they mention base J in the discussion without any explanation. It would be helpful to clarify certain points in the text and figures, e.g., it isn't clear why '3A' (and other subtelomeres) is found on multiple new contigs in the nanopore assembly. In Fig. 1A, it is hard to understand what the

lower panels represent, and how this corresponds to the upper panels, e.g., 5A and 5B appear to be adjacent in the Circos plot, but alternatives in the lower plot. In Fig. 2, the colours and transparencies used make it difficult to see anything other than the early S phase data. In Fig. 4, the colours in the violin plots are often impossible to distinguish and in panel A, it is unclear what is the relationship between *tig00652* and *tig00653* (are these homologues?). The three different grey scales used to annotate different repeats/telomeres aren't sufficiently distinct. In some cases, figures would be clearer if not represented as Circos plots, e.g., in Figs. 3 and S7, comparisons between samples is difficult for the reader. If the core and subtelomeric portions of the genome were represented on lines, with the data as curves, each sample could be vertically aligned to aid comparisons. Finally, it would aid the reader if the discussion included a few additional references to the main figures.

2. The genome assembly is an improvement on the previous best assembly, but it is still some way from a true telomere-to-telomere (T2T) assembly. This is a pity, since this is clearly the potential of nanopore sequencing. However, it is important to recognise the complexity of the *T. brucei* genome and the potential for variation between strains in different labs. Therefore, this new nanopore assembly is valuable, but also has limitations. There are some additional steps that the authors should take to add value, particularly if there is a way for the authors to estimate the completeness of their assembly. For example, what is the nanopore read coverage across different parts of their genome assembly, particularly repetitive regions, such as the centromeres? This will give an indication of whether some of the repetitive regions have collapsed in the assembly, which is likely given that the coverage on very long reads remains quite low. In fact, it would be useful for the authors to add to the supplementary table information on overall coverage on long reads, many studies using nanopore reads for T2T assemblies are limiting to reads >50 kb. In addition, the authors note that lack of completeness for their assembly could be due to a challenge associated with VSG-proximal sequences, potentially a 70 bp repeat, which they suggest may be problematic for nanopore sequencing. The authors could look for some evidence of this - for example, are the read quality scores of these 70 bp repeats lower than for other sequence contexts? Ultimately, this could be resolved by combining different nanopore chemistries, specifically by the addition of some R10 data. Do the authors have some R10 nanopore data that they could add and test whether this improves their assemblies?

3. The authors need to take care in their interpretation of their DNA replication datasets and in how they describe their data. Specifically, I have two concerns. First, the authors sometimes conflate early replication with a replication origin. For example, in the discussion the authors state "Centromeres are early replicating origins in yeast" - this is not correct. Almost all budding yeast centromeres replicate early but are not origins, they are replicated early from proximal origins 5-20 kb away. Another example, on pp 6 the authors state "MFA-seq revealed that centromeres located in the subtelomeres of chromosomes 9, 10 and 11 always displayed a peak, indicating they act as origins" - it would be clearer to state that this means that the centromeres are early replicating and must contain replication origins. Second, the authors need to be more careful in their data interpretation and recognise the limits of a population-level analyses. Peaks in MFA (sort-seq) data represent genomic loci present at higher copy number than other loci. This is consistent with a cell population average early replication time which implies that the region includes one or more replication origins. Regions that lack clear peaks are likely to still encompass replication origins, but these origins are probably variable in location between cells and consequently there are no clear peaks. The method can only detect differences in replication time that are consistently present within the population of cells, often called 'high efficiency' origins. For example, in the first paragraph of the discussion the authors state "subtelomeres are largely devoid of origins" - what they probably mean is that subtelomeres are devoid of MFA-seq peaks and therefore likely to be replicated from many low efficiency origins (high cell-to-cell variability). Their data also suggest that the replication of the subtelomeres will tend to be in late S phase - consistent with low efficiency origin sites. Finally, the authors state that "we show that the 177 bp repeats act as DNA replication origins" - I think this statement is too strong, the authors haven't really demonstrated this, it's just one model consistent with their data.

4. The data that the authors present is of high quality and generally quantitative. However, there are frequently qualitative statements in the manuscript that should be reported in more quantitative terms, and generally with an associated appropriate statistical test.

For example:

- pp 8: "In the subtelomeric sequence of this contig, mean BSF S/G2M ratios were notably lower than those observed within the BES1 transcribed region of both *tig 652* and *tig653* (Fig.4A)." This statement reports the observed difference, but should be quantified and the statistical significance of the difference tested.

- pp 12 (but also equivalent sections of the results): "Improved assembly of the subtelomeres now reveals the scale of this gene loss and shows that the extent of such instability is even greater in RAD51 mutants." These statements should be quantitative, and the significance tested.

- pp 7: why are the subtelomeres underrepresented in sequence coverage compared to the core genomes, e.g., comparing left and right panels of Fig S7? In this section of the manuscript, there are many qualitative statements that need to be quantitative and backed up by statistics.

5. The authors illustrate the value of a more complete assembly by reanalysing published short read datasets that are now mapped to the new assembly. However, there is an important potential caveat with this analysis that should be considered and/or discussed. Since the authors clearly state that previous short-read technologies weren't sufficient to unambiguously assemble through repetitive sequence, how can they be confident that remapping of short read data will be reliable in the repetitive regions of their new nanopore assembly. Many of the figures show short read data mapped across regions annotated as repetitive. The methods don't describe the parameters used for read mapping or filtering the mapped reads - how was this done? What criteria were used to ensure unique mapping of reads and to account for lower absolute levels of unique mapping within repetitive sequence? This is important, since it is a major stated advantage of the improved assembly, yet it's not clear how this advantage was achieved.

Minor comments:

1. In the introduction the authors state "MFA-seq; termed sort-seq in yeast" - this implies that they are equivalent terms. However, these are two different methods, please see this paper:

Müller, C. A. et al. The dynamics of genome replication using deep sequencing. *Nucleic Acids Research* 42, e3–e3 (2014). MFA, is a comparison of marker frequency between an exponentially growing cell population and a stationary phase population. Whereas sort-seq enriches for S phase cells by DNA content on a cell sorter (FACS), which are then compared to a non-replicating control sample that may also be acquired by cell sorting. At least in yeast, these methods are quite distinct with different utility.

2. In the discussion the authors state: "In addition, it seems likely that centromeric origins can overcome features, such as heterochromatin, that suppress transcription in the subtelomeres." - it would be appropriate for the authors the mention and cite that this is also the case in *S. pombe* where mechanisms have been elucidated by Susan Forsburg's group.
3. When discussing genome compartmentation, the authors make analogy to work in archaea but it would seem appropriate to also consider comparisons to work in mammalian cells, where A and B compartments have been described with clear links to replication domains, gene expression and chromatin marks.
4. It would be useful for the authors to define 'subtelomere', particularly in the context of the *T. brucei* genome.
5. In the first sentence of Introduction there is a small typo: "The fullest possible understanding of genome sequence is a critical resource to describe and analyse the biology of an organism, including of genome transmission and stability through DNA replication."

Reviewer #4

(Remarks to the Author)

Reviewer #5

(Remarks to the Author)

Version 1:

Reviewer comments:

Reviewer #1

(Remarks to the Author)

The authors have responded appropriately to all the comments and the manuscript is suitable for publication

Reviewer #2

(Remarks to the Author)

By providing additional data and clarification, the authors have addressed most of the points we raised, particularly regarding the number of putative genes in the new assembly.

However, there are still 2 points that are not clear to us:

1) Genome assembly / quality control of the assembly

We found it difficult to interpret the data provided in the response to the reviewers, and so it is still not clear to us why the authors are so confident that many of their assembled contigs actually contain parts of several different chromosomes. However, we know that a genome assembly is never 'finished' and we appreciate the authors' efforts to produce an improved assembly of the subtelomeric regions.

2) For the reasons outlined in our initial response, we remain unconvinced that the data presented support the statement that:

"the compartmentalisation of DNA replication between the cores and subtelomeres of the megabase chromosomes is associated with differing levels of stability in the two compartments and is influenced by homologous recombination."

Despite this disagreement, we believe that the manuscript contains several important findings and strongly support its publication in *Nature Communications*.

3) Data availability. We strongly believe that NGS data should be made available to reviewers, as a thorough evaluation of the data presented is not possible without the primary data. If the authors are unable to generate a "reviewer code", the data should be made public.

Minor:

Line 224: typo, "extent" should be "extend"

James Budzak and Nicolai Siegel

Reviewer #3

(Remarks to the Author)

The authors have done an excellent job on responding to comments and I recommend the manuscript for publication.

Reviewer #4

(Remarks to the Author)

Reviewer #5

(Remarks to the Author)

We are grateful to all five reviewers for their time and efforts in evaluating our submission to Nature Communications. We have now made a number of changes to the manuscript in response to the helpful comments.

The major changes are listed below, and the rationale for these changes and our detailed responses to the reviewers' comments are explained below (red text). In addition, associated text changes in the manuscript have been highlighted in red.

Thank you again for your evaluation, and best wishes,

Richard McCulloch

Summary of major changes:

Fig.2. Amended in response to reviewers #2/4 and 3/5 by increasing label and axis text size, and by highlighting the subtelomeric compartment of the chromosome contigs.

Fig.4. Amended in response to reviewers #3/5, improving labelling and adding statistics.

Fig.6. Telomere annotation added.

Table S1. Amended in response to reviewers#3/5, adding 50 kb+ read coverage statistics (median, mean, % covered etc.) across the assembly.

Table S2. We have simplified this table to remove estimates of total number of genes, as new analysis suggests the assembly contains some duplication. Total numbers of predicted *VSG* genes are retained, however, as this metric suggests good correspondence with the Muller genome. In addition, information on assembly gaps is removed.

Supplementary file 1. A new supplementary item, detailing 72 genes and pseudogenes predicted in the Nanopore genome assembly but not previously predicted in the Muller assembly.

Fig.S1. In response to reviewer #1, in each contig we had added mapping metrics of Nanopore reads >50 kb in length, allowing the reader to evaluate genome assembly quality.

Fig.S8. New figure, detailing read quality across the 70 bp and telomere repeats.

Responses to REVIEWER COMMENTS

Reviewer #1:

This manuscript describes the use of nanopore sequencing technology to dissect the dynamics of DNA replication in different genomic compartments of the *Trypanosoma brucei* genome. The work has revealed new insights' showing the replication of the active VSG expression site is directed from the telomere, that the core and subtelomeric regions display different levels of genomic stability which may relate to the use of subtelomeres for the immune evasion genes. They have demonstrated that the 177bp repeats act as replication origins and furthermore that these are found in the centromeres of the larger chromosomes too. The data support the conclusions drawn. This is a very carefully performed study which adds greatly to the understanding of the trypanosome genome and generates useful testable hypotheses.

Minor point: are the newly discovered hypothetical transcribed genes conserved across kinetoplastids or are they a *T. brucei* specific set of genes.

Thank you for this helpful suggestion. We have gone back and looked at the data and, when looking at hypothetical genes specifically, out of a total of 27 ortholog groups (107 genes) found on the sub-megabase chromosomes, 24 ortholog groups (90 genes) are *Trypanosoma*-specific; of these, 21 groups (78 genes) are only found in *T. brucei* and *T. evansi*.

We have reflected this further analysis by *adding the following to the text*:

'However, the largest group of genes were a diverse selection of hypotheticals (Fig.5D), the majority of which (24 out of 27 ortholog groups) appear to be specific to Trypanosoma species and suggest a wider range of encoded activities.'

I recommend acceptance of this manuscript.

Reviewers #2/4:

Trypanosoma brucei has a complex genome organisation, with most protein-coding genes located in transcribed chromosome cores while large archives of silent variant surface glycoprotein (VSG) antigen genes and VSG transcription units are located in arrays present in the heterozygous subtelomeric regions. In addition, the *T. brucei* genome contains several repetitive elements spanning many kilobases, including 70 bp repeats, 177 bp repeats, 50 bp repeats, centromeres and telomeres.

Despite significant advances in assembling a complete *T. brucei* genome, a telomere-to-telomere assembly has not been reported, meaning mapping of NGS data to these repetitive/unassembled regions cannot be accurately determined.

Here, Krasilnikova et al., use long-read Nanopore sequencing to assemble many of these repetitive elements and subtelomeric loci. The authors have previously mapped origins of replication as well as DNA replication dynamics using MFA-seq, however this analysis was mostly limited to the transcribed core regions as well as the VSG transcription units. The authors now provide a more complete picture of these replication dynamics by re-mapping previously published MFA-seq data to this new assembly.

As all papers from the McCulloch lab, this manuscript is very well written and contains a beautiful introduction and extensive discussion that nicely put the work in perspective.

Although I have some questions and concerns (see below), I believe the work presented here represents a very important step towards a proper understanding of genome replication in trypanosomes.

We thank the reviewers for their positive summary.

My main concerns relate to the accuracy evaluation of the new genome assembly and the claim that subtelomeric regions are more unstable than the chromosome cores.

Major points

1) Genome assembly:

a) Quality check of the assembly

Based on Fig. 2, contig Tig00004860 contains DNA from chr 1, chr 6 and chr 7, contig Tig00000002 contains DNA from the core of chr10 and subtelomeric DNA from both haplotypes 5A and 5B at the same time? Similarly, based on Fig1a the core of chr 4 (tig58) contains subtelomeric DNA from both haplotypes 5A and 5B at the same time.

It is very surprising that DNA sequences from different chromosomes are merged into one contig. If this is real, it would suggest that the original Müller was incorrectly scaffolded. How did the authors evaluate the correctness of the assembled contigs? If it was incorrectly assembled here, could it be that other contigs are also incorrectly assembled?

We hope that we did not give the impression that we consider the Muller assembly to be incorrect; instead, we suggested that the most likely reason for differences between the Nanopore assembly and the Muller genome result from known variation in the subtelomeres of these parasites (eg as described by Callegas et al Genome Res. 2006 Vol. 16 Issue 9 Pages 1109-1118):

*'Changes in predicted chromosome structural organisation between the Nanopore Lister 427 assembly here and the Müller PacBio-HiC genome⁹ may reflect limitations of assembly by either approach⁴⁸, or might reveal genuine differences between genome organisation in the stocks of strain between labs, which have been grown independently for decades. Since much of the differences are found in the subtelomeres, which are known to underlie differences in chromosome size between *T. brucei* strains⁸, the latter explanation may be favoured.'*

Nonetheless, to further address these concerns, we extracted ONT reads >50 kb in length that were used to assemble the genome and below we show their mapping, along with mapping quality, across tigs 2, 58 and 4860 (Figs1, S1 and S2), as examples of all the contigs. *In addition, we have modified Fig. S1 by adding 50 kb+ read mapping to all contigs >1 Mb to allow the reader to see coverage (and have changed the figure legend to reflect this addition).*

The mapping shows uninterrupted, overlapping coverage of long reads, with high mapping quality across the regions specified. In our view this is a positive indicator for assembly correctness, although misassemblies, as with any *de novo* genome assembly, cannot be excluded.

tig00000002 (5,080,222 bp)

Contig	Start	End	Reference contig	Start	End
tig00000002	11059	68413	Chr10_5A_Tb427v10	1570	58950
tig00000002	61763	73385	Chr10_5B_Tb427v10	59504	70979

tig00000058 (2,774,487 bp)

Contig	Start	End	Reference contig	Start	End
tig00000058	2371123	2421132	Chr4_5B_Tb427v10	33	50058
tig00000058	2622467	2707862	Chr4_5A_Tb427v10	3	85411

tig00004860 (4,554,032 bp)

Contig	Start	End	Reference contig	Start	End
tig00004860	88	764131	Chr1_3A_Tb427v10	1410584	2151564
tig00004860	788462	892549	Chr6_3A_Tb427v10	140125	244412
tig00004860	895282	2036867	Chr1_3A_Tb427v10	265722	1409561
tig00004860	2018380	2146992	Chr7_5A_Tb427v10	345171	474562
tig00004860	2143204	2532106	Chr7_5A_Tb427v10	474572	863519

0 to 2.6 Mb region spanning Chr1_3A, Chr6_3A, Chr7_5A and Chr7_core boundaries

Chr1_3A to Chr6_3A boundary:

tig00004860 (4,554,032 bp) continued

Contig	Start	End	Reference contig	Start	End
tig00004860	88	764131	Chr1_3A_Tb427v10	1410584	2151564
tig00004860	788462	892549	Chr6_3A_Tb427v10	140125	244412
tig00004860	895282	2036867	Chr1_3A_Tb427v10	265722	1409561
tig00004860	2018380	2146992	Chr7_5A_Tb427v10	345171	474562
tig00004860	2143204	2532106	Chr7_5A_Tb427v10	474572	863519

Chr6_3A to Chr1_3A boundary:

Chr1_3A to Chr7_5A to Chr7_core boundary:

The authors used Busco to evaluate the quality of the assembly, but I believe that it only assesses the completeness of the housekeeping gene assembly and does not yield information about the overall correctness of the genome assembly.

We agree, BUSCO only provides an indication of the completeness of the assembly in terms of presence of conserved and 'expected' eukaryotic genes, and from this we conclude considerable similarity to the Muller genome:

'Nonetheless, genome completeness measured using BUSCO³² indicated the Nanopore assembly was comparable to that of the Müller genome (Table S2)'

b) Table S2.

- The authors show that there are an additional 3708 protein-coding genes in their new assembly compared to the 427 2018 assembly. Where/what exactly are these genes?

In an early draft of the paper, we included a statement noting the possibility that the Nanopore assembly could contain duplications, but we removed it because we did not test this suggestion. Now, in order to address the reviewers' question, we performed pairwise protein BLAST comparisons between the putative protein sequences of the Muller assembly and the Nanopore assembly, using a range of minimum identity % values (see Table below).

	Muller 427 2018						ONT assembly					
Minimum identity threshold	98%	95%	80%	60%	50%	30%	98%	95%	80%	60%	50%	30%
Hits above threshold	104 51	107 63	111 29	112 70	113 17	113 81	121 73	133 28	157 88	178 04	183 96	191 78
Hits below threshold	934	622	256	115	68	4	705 0	589 5	343 5	141 9	827	45
Unique	3	3	3	3	3	3	72	72	72	72	72	72

Based on the above, 3708 is an overestimation of the number of 'additional' genes, likely due to the inclusion of phased genomic segments (ie duplication). We purposefully avoided collapsing/de-

duplicating the genome assembly to preserve as much of the variation between the hemizygous chromosomes as possible, and the counting of some genes more than once is an unfortunate consequence of this, which QAST doesn't appear to handle accordingly. The output of QAST mentioning 3708 'new' genes in the assembly is therefore misleading, *and we have removed it from the manuscript by simplifying the content of Table S2.*

Nonetheless, this analysis does suggest a much smaller number (72) of putative 'additional' genes revealed by the Nanopore assembly. In order to characterise the 72 genes (9 genes, 63 pseudogenes) that appear to be present in the Nanopore assembly, but not in the Muller genome, we used VEuPathDB's OrthoMCL pipeline to align the protein sequences to OrthoMCL orthogroups (using DIAMOND blastp) and found that 24/72 were assigned to a total of 23 orthogroups. In terms of their genomic location, all 72 are located on megabase chromosome contigs. Since these data may be of value, *we have generated a supplementary Excel file (Supplementary file 1) with this information, added the associated methods, and included the following text change to the results:*

'Only 72 putative genes and pseudogenes (Supplementary file 1) were predicted in the Nanopore assembly that were not detected in the Muller genome, and all were present in megabase chromosome contigs.'

- For the total length, is the sequence of the two homologous chromosomes counted separately?

The total length mentioned here is the total length of the assembly.

- In the Müller genome, Ns were used to indicate gaps in the scaffold. Thus, the statistic about the number of reduced Ns is not very informative.

With the exception of the last two rows, Table S2 is almost entirely the output of QAST, and the statistic for the number of N's is part of QAST's default output and was therefore included. We agree that the information does not add significant knowledge, and have removed this information from Table S2.

- I think that the differences in the number of contigs number is only partially informative. For example, in the Müller genome, cores and subtelomeres were separated for easier visualization of data. Also, the new nanopore assembly seems to have a much smaller number of contigs containing 177 bp repeats (8 compared to ~60 in the Müller genome. These are presumably contigs from intermediate or minichromosomes. Why are these contigs missing? In contrast, the new nanopore assembly seems to be much better at predicting the length of repeat regions especially, the 50 bp repeats seem to be much better covered in this assembly.

The total number of contigs in the Nanopore assembly that contain 177 bp repeats is 28; 10 of these are megabase chromosome contigs, 8 are what we have termed sub-megabase chromosome contigs, and 10 are too short to clearly categorize. In the Muller/Cosentino genomes, we can detect a total of 49 contigs containing 177 bp repeats; 7 are megabase chromosome contigs, while 42 are unitigs between 7.5 and 41 kb. Out of the 42 unitigs, 8 become incorporated into larger, sub-megabase contigs in the Nanopore assembly, and 2 become incorporated into >1 Mb contigs; the remaining 32

contigs contain a variable number of repeats, and in most cases 1-2 putative genes (median contig length 19.5 kb, mean 18.7 kb).

While we cannot provide a definitive explanation as to why 177 bp repeat-containing contigs are less abundant in our assembly, one possible explanation may lie in the apparent low base quality of telomeric repeat-containing Nanopore reads (discussed more fully below, in response to reviewer #3). Considering minichromosomes are very short, and more likely to be spanned by single long Nanopore reads, the low base quality observed for telomeric repeats may lead to most of long reads spanning these chromosomes being filtered out during the assembly process, leaving insufficient sequence for assembly of those chromosomes.

c) I find Fig 2 difficult to read. The labels are very small and it is not always clear where the boundaries between cores and subtelomeres are. As a consequence, the 'compartmentalization' is not clearly visible to me. Could the authors label the boundaries? Also, I don't find the contig label very useful. The 'compartmentalization' is easier to see in Fig S5. Perhaps the authors could merge the information from both figures and highlight where the new assembly has provided additional insight.

We agree that this figure was hard to read, and have therefore *adjusted the figure to incorporate the reviewers' suggestions*: the figure has been split over two pages so that the information becomes more legible; we have increased font size of contig labels and text in the legend box; and we have added grey boxes to the regions of the contigs that correspond with the subtelomeres in the Muller genome, thereby highlighting the different chromosome compartments and where the boundaries between them lie. I'm afraid that we decided to keep Fig. 2 and Fig. S5 separate, as merging the two would result in a very crowded and most likely illegible figure; in addition, we wish to allow the reader to be able to examine the MFA-seq data across the >1 Mb Nanopore contigs, which are contiguous sequences and are compared in a number of figures (Figs. 1, S1, S2, S3, S4). We acknowledge that the contig label is large and not *per se* informative, but this is also necessary to allow the reader to make connections across these figures, as well as to find information on the different contigs in the various files containing the genome data and information.

The figure legend has been amended to incorporate the changes.

d) Line 316. The authors write "These data suggest that the Nanopore assembly has revealed a potentially novel class of sub-megabase chromosomes,...". What makes the authors think that this is novel class of chromosomes and not simply intermediate chromosomes? However, it is very interesting that the authors identify actively transcribed regions within these chromosomes. It is important that the authors clarify if they are filtering out multimapping reads or not. If all reads are used, it could be that the reads are from another genomic locus.

As the reviewers have noted, the novelty of what we term sub-megabase chromosomes lies in the fact that each of these contain regions containing transcribed genes. These genes are not merely VSGs or ESAGs, which are the only genes routinely described on mini- and intermediate chromosomes and should either be transcriptionally silent (VSGs in minichromosomes) or subject to monoallelic expression (ESAGs and VSGs in BEs within intermediate chromosomes). Whether the contigs represent a truly distinct chromosome type, is unclear, but the presence of 177 bp repeats –

a 'signature' of mini- and intermediate-chromosomes (Wickstead, B., et al. (2004). "The small chromosomes of *Trypanosoma brucei* involved in antigenic variation are constructed around repetitive palindromes." *Genome Res* 14(6): 1014-1024.) - supports our suggestion (discussion) that there is a broader range of submegabase chromosomes than has been described to date.

Regarding RNAseq mapping: as mentioned in the Methods section, we only mapped uniquely mapping reads – i.e. those that only map once in the genome. *We have reworded the methods to emphasise this point:*

'RNAseq data was trimmed using trim galore as above, aligned using hisat2 (v.2.2.1) (settings: --no-spliced-alignment)104, with further processing to sorted bam file as above, *with additional samtools filtering retaining only reads that map once in the genome (samtools view -Sbu -d NH:1 file.sam > file.bam).*'

e) Line 241 What do the authors mean by 3 BES were duplicated? Do they mean that they are duplicated in the genome or that they have multiple contigs covering parts of the BESs?

To address this question, we extracted Nanopore reads >50 kb in length and investigated their mapping across the six BES in question. Based on the lack of reciprocal mapping of long reads between 'duplicated' BES sequences, we propose that these are likely to be distinct copies of a BES in all three cases. As the cells used for Nanopore sequencing were not subcloned for the purpose of sequencing, we are unable to determine whether the duplications are likely to be present in the same cell or are simply a consequence of a heterogeneous cell population and frequent rearrangements in the subtelomeres.

2) Instability of subtelomeric regions

Line 234: Instability of subtelomeric regions

I am not convinced that the data really demonstrate higher instability in subtelomeres compared to cores in WT. Perhaps I am missing something, but I only see a decrease in coverage in one clone in Fig 3, nothing in Fig S7. For the RAD51-null mutant there are clear differences. However, a difference in coverage does not really mean that there is lower stability, the subtelomeres are haploid, thus any loss will be much more visible. In addition, any loss in the core region would be likely associated with a severe growth defect and would therefore be selected against.

*We acknowledge these limitations, and indeed have amended the text to make them more clear (see below), but respectfully we disagree, for several reasons. First, in the timecourse experiment (Fig.3) there is indeed limited evidence for decreased coverage that arises over the short period of growth in wild type cells, but in the starting population (Fig.S7) there are multiple regions with lowered coverage in the subtelomeres that are retained during the timecourse, arguing for greater levels of ongoing sequence loss than in the cores. Second, these data are valuable as they extend significantly on previous work that has detailed instability in *T. brucei* BRCA2 and MRE11 mutants, but was limited to Southern-based analysis of select VSGs (Trenaman et al. *Nucleic Acids Res* 2013 Vol. 41 Issue 2 Pages 943-60; Robinson et al. (2002) *J.Biol.Chem.* 277(29): 26185-26193); the data we provide here show the scale of gene loss in the subtelomeres, and reinforce the suggestion of ongoing*

recombination in this genome compartment. Finally, these data add to previous work in archaea that reveals an association between origin location/density and genome (in)stability, suggesting a generalised phenomenon. *Text added to the discussion:*

'Nonetheless, it may be that compartmentalisation of the T. brucei genome into a highly transcribed core of mainly 'housekeeping' genes and a subtelomere compartment that mainly houses virulence genes (VSGs), means that gene loss in the latter is more readily tolerated than the former.'

3) 177 bp repeats as origins of replication.

a) The origins of replication described in the 177bp repeats are intriguing, but the only experimental evidence the authors provide that these act as origins is MFA-seq. It would be nice to see CHIP-seq against an ORC protein (or other similar DNA replication protein) to see if the MFA-seq matches ORC binding in these regions. The authors could map the available data. Also, can the authors really rule out the possibility that the telomeric sequence is acting as an origin of replication?

The reviewers are correct as regards the limitation of stating the 177 bp are origins, and this concern is shared with reviewers #3/5. In acknowledgement of the lack of further evidence, *we have modified the text in several places to state these are putative origins.*

Key changes:

Abstract: 'Lastly, *we provide evidence that* the 177 bp repeats act as widespread, conserved DNA replication origins'

Section title: '177 bp repeats *may be* widely conserved DNA replication origins in T. brucei'

Discussion:

'the transcribed core of the megabase chromosomes is stably maintained by multiple replication origins, many of which share 177 bp repeats *where DNA replication initiates* in the sub-megabase chromosomes';

'Furthermore, *we provide evidence that* the 177 bp repeats act as DNA replication origins';

'The *suggestion that* the 177 bp repeats are a previously undetected, widely conserved DNA replication origin...'

We are unfortunately not able to add ORC mapping data, as this was run as a CHIP-chip experiment and repetitive sequences in the genome, including 177 bp repeats were excluded.

The reviewers bring up a fair concern that the higher MFA-seq ratios we detect over the 177 bp repeats might not stem from the repeats themselves but from the telomeres. We have inspected the contigs in Fig.6 and, of the six, two have telomeric repeats – tig00004879 (telomere downstream of the 177 bp repeats) and tig00000194 (telomere at the opposite end of the contig to the 177 bp repeats). By error, the telomeres were not highlighted the previous version of Fig. 6; *this has now been amended.* As we document in the manuscript, MFA-seq ratios at telomeric repeats in all contigs are ~1.5 in early S samples of BSF cells, but not in PCF cells. In tig00004879, where the 177 bp repeats and telomere are adjacent, we see MFA-seq ratios ~1.5 in both BSF and PCF cells, and hence here we cannot separate replication activity at the two repeat types. However, in tig00000194,

where we detect MFA-seq ratios of ~1.5 around the short stretch of 177 bp repeats in both cell types, we see ratios of ~1.5 at the distal telomere repeats only in BSF cells, not PCF. Thus, here, we see evidence for the 177 bp repeats, and not the telomeres, acting as sites of replication initiation in PCF cells. Nonetheless, as we stress in the discussion, definitive demonstration that the 177 bp repeats act as DNA replication origins needs to be tested experimentally (in the future):

'For instance, it will be interesting to test if the 177 bp repeats provide a conserved sequence feature for ORC binding^{52,86}, thereby dictating origin activity.'

b) It would be nice to see more information on possible mechanisms of how repetitive regions of the genome can act as of origins of replication in other organisms and in trypanosomes (either in the introduction or in the discussion).

We have discussed and cited papers that describe ORC function at telomeres in other eukaryotes, as well as the limited information that is available about ORC at telomeres in *T. brucei* (discussion):

'ORC has been described to interact with telomeres in other eukaryotes^{87,88}, and loss of T. brucei ORC impairs telomere integrity and leads to altered VSG expression⁸⁹.'

c) Line 264. The authors report a decrease in MFA-seq reads in the subtelomeres. To better see this drop, it would be good to plot the data next to each other.

We have improved presentation and analysis of Figs.4 and 6, and hope that these changes make this clearer.

d) Line 274. Did the authors do any filtering for mapping quality (uniqueness of reads)? I assume that the drop in read coverage over the 50 bp is caused by the authors using only uniquely mapping reads. If this is the case, how were the authors able to analyse coverage across telomeric repeats?

We apologise for not being clear on this point in the main text or methods. *A more detailed description of the analysis has now been added in the methods*, where it is made clear that a MapQ of 1 was used, as well as filtering for a minimum number of reads that map (below).

To address the reviewers' comment on the lack of signal across some parts of the 50 bp repeat regions, we looked more closely at the data. Prior to calculating the MFA-seq ratio (replicating sample vs non-replicating sample), two layers of filtering are applied: a MapQ of 1, and a minimum of 100 reads per 1 kb bin. Looking at the file containing all the excluded bins across 50 bp regions shows that there are reads mapped across these regions (after MapQ of 1 is applied), but that these were excluded from downstream analysis because they contained < 100 reads (graph shown below); the only exception was tig55, where none of the bins across the 50 bp repeat region were excluded.

Regarding the telomeric repeats, none of the bins in these regions were filtered out as all contained > 100 reads. The reviewers raise a valid concern on how we trust reads mapped to the telomeric repeats regions. We have confidence in these results for two reasons: first, although multi-mapping is taking place, we assume this will be equivalent across all samples, and the MFA-seq is a ratio where we normalise in one sample relative to G2/M; second, we see clear differences between PCF and BSF life cycle stages (Fig. 4), with evidence of replication (MFA-seq ratio>1) only in BSF cells, suggesting that this is a biological, rather than a technical, difference.

e) Line 284. I do not understand how the authors can conclude that there is general telomere-initiated DNA replication activity in BSF. Couldn't the signal come from the active BES? Or from early replicating MC? This problem is acknowledged by the authors in the discussion but it should be considered earlier.

With respect, we feel that this is best covered in the discussion, as it is an interpretation, not a description of data.

f) Line 29. The authors write “To date, genome-wide mapping of DNA replication has not included intermediate- and mini-chromosomes due to lack of assembly in the strain for which MFA-seq data is available”. However, Cosentino et al (2021) describe the identification of 60 contigs with 177 bp repeats for the Lister 427 strain. Why did authors not map their MFA-seq data to these contigs?

For the purpose of analysing the MFAseq data, we limited our analysis only to the newly assembled sub-megabase chromosome contigs shown in Figs. 5 and 6, as these provide sufficient resolution and genomic context for clear analysis. To illustrate the difficulty of performing the same analysis is putative minichromosomes, below we have plotted MFA-seq data across the three longest Muller/Cosentino sequences that contain 177bp repeats but are not part of megabase or larger sub-megabase chromosomes. Here, due to their small size, there is very sparse data points, and MFA-seq ratios >1 are essentially seen across the entire contigs, since most of the content in 177 bp repeats. As a small extra note, these data reflect what is seen in Figs.5 and 6, in that ES/G2M and LS/G2M ratios are more clearly higher than G1/G2M ratios in BSF cells than in PCF cells. As we do not have an explanation for this difference, which is not seen in the megabase chromosomes, we have chosen not to discuss it in the submission. .

4) The sequencing data is not accessible at EBI under the accession number PRJEB75456.

The data are present and will be made public once this work is published; were not able to generate a reviewer code.

Minor points

Line 56: The authors state that the parasite has genome of 60 Mb and cite Berriman et al. However, this reference states that the genome is 26 Mb in size. Are the authors referring to the combined size of both homologous chromosomes?

We have clarified this: 'The parasite's modest ~60 Mb (*diploid*) genome...'

Line 37-39: This sentence is slightly misleading. The 177bp repeats are present on minichromosomes and intermediate chromosomes, however VSG transcription sites are only located on intermediate chromosomes and not minichromosomes. This is not made clear here, either the sentence should be reworded or a distinction should be made between these chromosome types.

Corrected:

'and multiple VSG transcription sites localise to the telomeres of *megabase and some submegabase chromosomes*'.

Line 105-107: this sentence did not read so well for me. I would probably rephrase

I'm afraid we cannot see how to improve this.

In Fig S1. there is a formatting error in the number displayed on the bottom left hand side of each chromosome schematic.

We have re-plotted Fig. S1 to also include > 50kb read mapping, and this has also resolved the formatting issue.

Line 599: BESS should be BESs

Corrected

Line 625: Should this be “Telomere-directed initiation” instead of “Telomere-directed the initiation”?

Thank you, corrected.

All instances of “HiC” should be “Hi-C”

Corrected.

Reviewers #3/5:

This manuscript leverages nanopore long reads to improve the genome assembly for *Trypanosoma brucei*. The authors then demonstrate the utility of the improved assembly by re-mapping previously acquired genomic datasets to gain new insights into trypanosome biology. The *T. brucei* genome is particularly complex (and distinct from most other eukaryotes), with long poly-cistronic transcription units, complex repetitive sequences and many smaller (poorly characterised) chromosomes. One crucial biological aspect of *T. brucei* is the ability to switch the variant surface glycoproteins (VSGs) to avoid the host immune response. There remains much to be learnt about the mechanisms that participate in the switching of the expressed VSGs, and this manuscript makes some valuable contributions in this area.

Overall, the manuscript feels a little like a somewhat uncomfortable combination of two stories. First, the improved genome assembly that whilst it could be seen as incremental has the potential to be a value to the research community. Second, a new characterisation of replication dynamics using existing datasets now mapped to the new assembly. Ultimately there is value in this combination, since it allows the authors to demonstrate the importance of improved genome assemblies for gaining greater insight into fundamental biological processes.

However, we feel that there could be some improvement to the structure of manuscript to help make it more accessible to a wider audience. The submitted version switches back and forth between describing new aspects of the assembly and new insights from re-mapped data. It would probably be clearer and more accessible, if the manuscript cleanly described the new assembly (e.g. the main chromosomes and novel genes identified on the smaller chromosomes) in an initial section, and then moved onto the more functional analyses that rely upon the remapping of existing datasets. Subject to this comment and those below, we feel that this is an interesting manuscript deserving of publication.

We thank the reviewers for their positive summary.

Major comments:

1. The manuscript would benefit from increased clarity of language and some simplification to figures. This would help make the manuscript more accessible to a wider audience. We recognise that the *T.brucei* genome is complex in nature and function, differing from most model species. We suggest carefully introducing the key differences between the *T.brucei* genome and common models avoiding wherever possible complex terminology, e.g. they mention base J in the discussion without any explanation.

We trust that the changes made to the figures (listed above and explained throughout this response) improve clarity, and we thank the reviewers for their suggestions. We are aware that we rush over describing the new assembly, and instead focus on DNA replication, but this is deliberate, as we do not wish to give the impression that this project provides a new genome assembly. This is for two reasons: as noted below, it is not a true telomere-to-telomere assembly; and, as we state above (reviewers #2/4), there is already a very good assembly for *T. brucei* Lister 427 and so we have focused on a small number of features we have improved understanding of using Nanopore: greater assembly of and across repetitive regions (in particular centromeres and 50 bp repeats); and assembly of overlooked submegabase chromosomes, much of whose content is 177 bp repeats.

It would be helpful to clarify certain points in the text and figures, e.g., it isn't clear why '3A' (and other subtelomeres) is found on multiple new contigs in the nanopore assembly.

Please see response to reviewers #2/4: we suspect these subtelomere changes reflect dynamic reassortment in this compartment of the genome, and this is explained in the discussion.

In Fig. 1A, it is hard to understand what the lower panels represent, and how this corresponds to the upper panels, e.g., 5A and 5B appear to be adjacent in the Circos plot, but alternatives in the lower plot.

We have improved description of this in the legend:

'for clarity of comparison, the organisation of the transcribed core, transcriptionally silent subtelomeres (numbered 3A, 3B, 5A, 5B) and bloodstream VSG expression sites (BESs) in these chromosomes in the Muller genome assembly are diagrammed in the lower panel (adapted from ⁹).

In Fig. 2, the colours and transparencies used make it difficult to see anything other than the early S phase data.

Please see the response to reviewers#2/4; we have substantially changed this figure to improve clarity.

In Fig. 4, the colours in the violin plots are often impossible to distinguish and in panel A, it is unclear what is the relationship between tig00652 and tig00653 (are these homologues?). The three different grey scales used to annotate different repeats/telomeres aren't sufficiently distinct.

We have re-made Fig. 4 to incorporate the reviewers' helpful feedback. We have changed highlighting of the 70 bp repeats (grey to green), added labels to the x-axis of the violin plots, and added statistical tests (see below). The figure legend has been modified to incorporate the changes. Both tig652 and tig653 incorporate BES1; whether this is duplicated in the genome of the cell line or

cell population we have sequenced, or if these are generated as separate contigs because of the genome assembly package used, is unclear (but we suspect the former; see above, reviewers #2/4).

In some cases, figures would be clearer if not represented as Circos plots, e.g., in Figs. 3 and S7, comparisons between samples is difficult for the reader. If the core and subtelomeric portions of the genome were represented on lines, with the data as curves, each sample could be vertically aligned to aid comparisons.

We have tried various formats of presenting these data and, in our view, Circos plots are clearest.

Finally, it would aid the reader if the discussion included a few additional references to the main figures.

We have done so.

2. The genome assembly is an improvement on the previous best assembly, but it is still some way from a true telomere-to-telomere (T2T) assembly. This is a pity, since this is clearly the potential of nanopore sequencing. However, it is important to recognise the complexity of the T.brucei genome and the potential for variation between strains in different labs. Therefore, this new nanopore assembly is valuable, but also has limitations. There are some additional steps that the authors should take to add value, particularly if there is a way for the authors to estimate the completeness of their assembly. For example, what is the nanopore read coverage across different parts of their genome assembly, particularly repetitive regions, such as the centromeres?

This will give an indication of whether some of the repetitive regions have collapsed in the assembly, which is likely given that the coverage on very long reads remains quite low. In fact, it would be useful for the authors to add to the supplementary table information on overall coverage on long reads, many studies using nanopore reads for T2T assemblies are limiting to reads >50 kb.

To address this question, and as discussed above (reviewers#2/4), we have added a read mapping track of all reads >50 kb to all contigs in Fig. S1. In addition, we added 50kb+ read coverage statistics (median, mean, % covered etc.) across the assembly to Table S1.

In addition, the authors note that lack of completeness for their assembly could be due to a challenge associated with VSG-proximal sequences, potentially a 70 bp repeat, which they suggest may be problematic for nanopore sequencing. The authors could look for some evidence of this - for example, are the read quality scores of these 70 bp repeats lower than for other sequence contexts? Ultimately, this could be resolved by combining different nanopore chemistries, specifically by the addition of some R10 data. Do the authors have some R10 nanopore data that they could add and test whether this improves their assemblies?

This has proved a very valuable suggestion indeed, so, thank you!

As suggested by the reviewer, we analysed base quality of reads spanning 70 bp repeat regions to identify potential sequencing issues; we did this by extracting and plotting 10kb+ reads along with their coverage and median base quality across 70 bp regions and flanking sequences (and we have provided this analysis as a new supplementary figure, Fig. S8).

Interestingly, base qualities across 70 bp regions didn't show an obvious drop, even with reduced coverage. However, we noticed that for the contigs that contain downstream telomeric repeats, the base quality falls noticeably. We applied the same approach to non-BES telomeric repeat regions and found the same pattern: reduced base quality at telomeric repeats. We therefore suspect that the reduced base quality of telomeric repeats may have led to telomere-containing reads to either be clipped or discarded by the assembly process, thereby reducing the coverage of telomere-proximal sequences, including BES VSGs at are immediately adjacent to the telomere.

We have added the following text:

'Analysis of the base quality of reads spanning the 70 bp repeats did not suggest they are especially problematic for the Nanopore sequencing process itself, but any sequences that contain telomere repeats were of notably lower quality (Fig.S8). Thus, it may be that features of the T. brucei telomere impact Nanopore sequencing, and thereby impede assembly of sequences in their proximity, including BESs.'

While we cannot determine the reason for the base quality drop in *T. brucei*, we note that this issue has been previously observed (doi.org/10.1186/s13059-022-02751-6; doi.org/10.1016/j.xgen.2024.100588). We can speculate that the base quality drop in this scenario may be due to the widely-known difficulty of ONT sequencing in handling homopolymer sequences (which the telomeric repeat, TTAGGG, would fall under) or, alternatively, a base modification at telomeric repeats – notably, it is known that *T. brucei* telomeric repeats are the main genomic location of the novel modified thymidine base, termed base J ([doi: 10.1073/pnas.95.5.2366](https://doi.org/10.1073/pnas.95.5.2366)), which may alter the ion current and therefore lower the base quality.

Regarding using more/newer Nanopore data for improved genome assembly, we had attempted this earlier this year, using more up-to-date dorado basecalling and additional sequencing runs (using a total of >7 gigabases of long read sequence), but this did not yield improvement in BES contiguity, unfortunately.

3. The authors need to take care in their interpretation of their DNA replication datasets and in how they describe their data. Specifically, I have two concerns. First, the authors sometimes conflate early replication with a replication origin. For example, in the discussion the authors state "Centromeres are early replicating origins in yeast" - this is not correct. Almost all budding yeast centromeres replicate early but are not origins, they are replicated early from proximal origins 5-20 kb away.

*We acknowledge that we have not been careful enough and have **changed the wording in several locations** (see response to reviewers #2/4 for origin changes, as key examples).*

Another example, on pp 6 the authors state "MFA-seq revealed that centromeres located in the subtelomeres of chromosomes 9, 10 and 11 always displayed a peak, indicating they act as origins" - it would be clearer to state that this means that the centromeres are early replicating and must contain replication origins.

Thank you for this suggestion, which we have incorporated.

Second, the authors need to be more careful in their data interpretation and recognise the limits of a population-level analyses. Peaks in MFA (sort-seq) data represent genomic loci present at higher

copy number than other loci. This is consistent with a cell population average early replication time which implies that the region includes one or more replication origins. Regions that lack clear peaks are likely to still encompass replication origins, but these origins are probably variable in location between cells and consequently there are no clear peaks. The method can only detect differences in replication time that are consistently present within the population of cells, often called 'high efficiency' origins. For example, in the first paragraph of the discussion the authors state "subtelomeres are largely devoid of origins" - what they probably mean is that subtelomeres are devoid of MFA-seq peaks and therefore likely to be replicated from many low efficiency origins (high cell-to-cell variability).

We again acknowledge the reviewers' concern *and have altered the text*, but we suggest that it remains possible that there are no origins in this compartment of the genome and have not altered discussion of this possibility:

'subtelomeres are largely devoid of origins that can be detected by MFA-seq'.

Their data also suggest that the replication of the subtelomeres will tend to be in late S phase - consistent with low efficiency origin sites. Finally, the authors state that "we show that the 177 bp repeats act as DNA replication origins" - I think this statement is too strong, the authors haven't really demonstrated this, it's just one model consistent with their data.

We have corrected the text

4. The data that the authors present is of high quality and generally quantitative. However, there are frequently qualitative statements in the manuscript that should be reported in more quantitative terms, and generally with an associated appropriate statistical test.

For example:

- pp 8: "In the subtelomeric sequence of this contig, mean BSF S/G2M ratios were notably lower than those observed within the BES1 transcribed region of both tig 652 and tig653 (Fig.4A)." This statement reports the observed difference, but should be quantified and the statistical significance of the difference tested.

- pp 12 (but also equivalent sections of the results): "Improved assembly of the subtelomeres now reveals the scale of this gene loss and shows that the extent of such instability is even greater in RAD51 mutants." These statements should be quantitative, and the significance tested.

Statistical analysis has been applied to the data represented as violin plots in Fig. 4. The figure legend (see above) as well as the main text have been adapted accordingly. The details of the statistical analysis have also been added to the Methods section – see reply to reviewers #2.

- pp 7: why are the subtelomeres underrepresented in sequence coverage compared to the core genomes, e.g., comparing left and right panels of Fig S7? In this section of the manuscript, there are many qualitative statements that need to be quantitative and backed up by statistics.

*Due to the hemizygous nature of *T. brucei* megabase chromosomes, the subtelomeric sequences are haploid, while the core genome compartments are diploid (collapsed) - this results in lower read depth coverage at subtelomeric regions.*

5. The authors illustrate the value of a more complete assembly by reanalysing published short read datasets that are now mapped to the new assembly. However, there is an important potential caveat with this analysis that should be considered and/or discussed. Since the authors clearly state that previous short-read technologies weren't sufficient to unambiguously assemble through repetitive sequence, how can they be confident that remapping of short read data will be reliable in the repetitive regions of their new nanopore assembly. Many of the figures show short read data mapped across regions annotated as repetitive. The methods don't describe the parameters used for read mapping or filtering the mapped reads - how was this done? What criteria were used to ensure unique mapping of reads and to account for lower absolute levels of unique mapping within repetitive sequence? This is important, since it is a major stated advantage of the improved assembly, yet it's not clear how this advantage was achieved.

For Illumina RNAseq data, only uniquely mapping reads were used (reads that map only once in a genome), as mentioned in the methods. Both DRIP-seq (R-loop) and MFA-seq datasets used minimum mapping quality 1 as standard, but, more importantly, both of these datasets are normalised – DRIP-seq is normalised against an input sample, whereas for MFA-seq analysis, the G2M samples are used for normalisation. *We have expanded the Methods section for MFA-seq to include more detail regarding normalisation, filtering and MAPQ values.*

Minor comments:

1. In the introduction the authors state "MFA-seq; termed sort-seq in yeast" - this implies that they are equivalent terms. However, these are two different methods, please see this paper:

Müller, C. A. et al. The dynamics of genome replication using deep sequencing. *Nucleic Acids Research* 42, e3–e3 (2014).

MFA, is a comparison of marker frequency between an exponentially growing cell population and a stationary phase population. Whereas sort-seq enriches for S phase cells by DNA content on a cell sorter (FACS), which are then compared to a non-replicating control sample that may also be acquired by cell sorting. At least in yeast, these methods are quite distinct with different utility.

*This is a historical mistake by us, and we are unfortunately stuck with the (incorrect) name: MFA-seq in *T. brucei* was, indeed, first run by FACS sorting, but later iterations in *Leishmania* are true MFA-seq, using a stationary phase population.*

2. In the discussion the authors state: "In addition, it seems likely that centromeric origins can overcome features, such as heterochromatin, that suppress transcription in the subtelomeres." - it would be appropriate for the authors the mention and cite that this is also the case in *S. pombe* where mechanisms have been elucidated by Susan Forsburg's group.

Thank you for this reminder; we have cited a recent review by Forsburg and Shen.

3. When discussing genome compartmentation, the authors make analogy to work in archaea but it would seem appropriate to also consider comparisons to work in mammalian cells, where A and B

compartments have been described with clear links to replication domains, gene expression and chromatin marks.

We have not cited these mammalian studies as it is not at all clear that replication domains operate in *T. brucei*, and so the comparison has likely limited value.

4. It would be useful for the authors to define 'subtelomere', particularly in the context of the *T. brucei* genome.

We have defined this in the abstract.

5. In the first sentence of Introduction there is a small typo: "The fullest possible understanding of genome sequence is a critical resource to describe and analyse the biology of an organism, including of genome transmission and stability through DNA replication."

Corrected

REVIEWERS' COMMENTS

Reviewer #1 (Remarks to the Author):

The authors have responded appropriately to all the comments and the manuscript is suitable for publication

Thank you for your response and consideration of our manuscript.

Reviewer #2 (Remarks to the Author):

By providing additional data and clarification, the authors have addressed most of the points we raised, particularly regarding the number of putative genes in the new assembly.

However, there are still 2 points that are not clear to us:

1) Genome assembly / quality control of the assembly

We found it difficult to interpret the data provided in the response to the reviewers, and so it is still not clear to us why the authors are so confident that many of their assembled contigs actually contain parts of several different chromosomes.

However, we know that a genome assembly is never 'finished' and we appreciate the authors' efforts to produce an improved assembly of the subtelomeric regions.

We agree that no assembly is ever complete, and please note that we do not claim this assembly is definitive.

2) For the reasons outlined in our initial response, we remain unconvinced that the data presented support the statement that:

“the compartmentalisation of DNA replication between the cores and subtelomeres of the megabase chromosomes is associated with differing levels of stability in the two compartments and is influenced by homologous recombination.”

Despite this disagreement, we believe that the manuscript contains several important findings and strongly support its publication in Nature Communications.

We accept the reviewers' concerns but stand by our data and assertions in the paper.

3) Data availability. We strongly believe that NGS data should be made available to reviewers, as a thorough evaluation of the data presented is not possible without the primary data. If the authors are unable to generate a "reviewer code", the data should be made public.

All data is public. See email confirmation:

From: NLM Support <nlm-support@nlm.nih.gov>

Sent: 27 November 2024 18:00

To: Marija Krasilnikova <Marija.Krasilnikova@glasgow.ac.uk>

Cc: marija.krasilnikova@gmail.com <marija.krasilnikova@gmail.com>; Richard McCulloch <Richard.McCulloch@glasgow.ac.uk>

Subject: RE: Case # CAS-1402348-X8S7R4: [EXTERNAL] PRJNA962304: release of unpublished data TRACKING:000435001405521

This SRA data is now public. Please allow several hours for this data to become indexed and searchable.

The SRA Team

National Center for Biotechnology Information | National Library of Medicine

National Institutes of Health, Bethesda, Maryland 20892

Email: SRA@ncbi.nlm.nih.gov

Minor:

Line 224: typo, "extent" should be "extend

Corrected

James Budzak and Nicolai Siegel

Reviewer #3 (Remarks to the Author):

The authors have done an excellent job on responding to comments and I recommend the manuscript for publication.

Thank you for your response and consideration of our manuscript.